

# Robust uncertainty assessment of the spatio-temporal transferability of glacier mass and energy balance models

Tobias Zolles[1,2,3], Fabien Maussion[1], Stephan Peter Galos[1], Wolfgang Gurgiser[1], and Lindsey Nicholson[1]

[1]Department of Atmospheric and Cryospheric Sciences, Universität Innsbruck, Innsbruck, Austria
[2]Institute for Geosciences, University of Bergen, Bergen, Norway
[3]Bjerknes Center for Climate Research, Bergen, Norway

*Correspondence to:* Tobias Zolles (tobias.zolles@uib.no)

**Abstract.**

Energy and mass balance modeling of glaciers is a key tool for climate impact studies of future glacier behaviour. By incorporating many of the physical processes responsible for surface accumulation and ablation, they offer more insight than simpler statistical models and are believed to suffer less from problems of stationarity when applied under changing climate conditions. However, this view is challenged by the widespread use of parameterizations for some physical processes introduces a statistical calibration step.

We argue that the reported uncertainty in modelled mass balance (and associated energy flux components) are likely to be understated in modelling studies that do not use spatio-temporal cross-validation and use a single performance measure for model optimization. To demonstrate the importance of these principles, we present a rigorous sensitivity and uncertainty assessment workflow applied to a modelling study of two glaciers in the European Alps.

The procedure begins with a reduction of the model parameter space using a global sensitivity assessment that identifies the parameters to which the model responds most sensitively. We find that the model sensitivity to individual parameters varies considerably in space and time, indicating that a single stated model sensitivity value is unlikely to be realistic. The model is most sensitive to parameters related to snow albedo and vertical gradients of the meteorological forcing data.

We then apply a Monte Carlo multi-objective optimization based on three performance measures: Model bias and mean absolute deviation in the upper and lower glacier parts, with glaciological mass balance data measured at individual stake locations used as reference. This procedure generates an ensemble of optimal parameter solutions which are equally valid. The range of parameters associated with these ensemble members are used to estimate the cross-validated uncertainty of the model output and computed energy components. The parameter values for the optimal solutions vary widely, and considering longer calibration periods does not systematically result in more constrained parameter choices. The resulting mass balance uncertainties reach up to 1300 $\mathrm{kgm}^{-2}$, with the spatial and temporal transfer errors having the same order of magnitude. The uncertainty of surface energy flux components over the ensemble at the point scale reached up to 50 % of the computed flux. The largest absolute uncertainties originate from the short-wave radiation and the albedo parametrizations, followed by the turbulent fluxes. Our study highlights the need for due caution, and realistic error quantification when applying such models to



regional glacier modelling efforts, or for projections of glacier mass balance in climate settings that are substantially different from the conditions in which the model was optimized.

# 1 Introduction

Surface energy and mass balance models are valuable tools for estimating the response of glaciers to meteorological forcing
(Oerlemans, 2011). Model results can be used to estimate regional run-off and resultant sea level rise (e.g., Hock, 2005), but additionally, and unlike results of empirical melt models, they can also be used to characterize the fundamental processes and key drivers of melt on glaciers, which is critical for understanding how they may behave under the influence of changing climate (e.g., Mölg and Hardy, 2004; Klok and Oerlemans, 2004; Hock and Holmgren, 2005; Mölg et al., 2008; Prinz et al., 2016; Willeit and Ganopolski, 2017).

All glacier surface mass and energy-balance models contain a degree of parametrization of physical relationships. These parameters are either optimized to fit measured glacier mass balance, or chosen based on previously established empirical relationships, or are a mix thereof. Uncertainty surrounding the transferability of parametrizations in both space and time poses a critical limitation on the usefulness of such models for regional upscaling of glacier behaviour or forward projections of global glacier behaviour under changing climate conditions.

Early energy balance studies typically apply models at a single point in space for which local physical relations can be readily established empirically, or direct measurements are available to tune the parametrizations (e.g. Mölg and Hardy, 2004; Greuell and Konzelmann, 1994; Bintanja and Van Den Broeke, 1995). Optimizing a model to local measurements can successfully reproduce local melt rates (e.g., Oerlemans and Knapp, 1998), and, where this is the case, reliable simulation of glacier ablation is often taken to mean that the model also accurately reveals the relative importance of specific energy sources to ice ablation.
Model optimization based on data from a single site, or from a very short time series, is, however, prone to parameter over-fitting, meaning that parameters are specifically adjusted to the study location and/or time (Beven, 1989). This can be evident in upscaling point optimizations to the glacier scale: For example, Klok and Oerlemans (2002) applied a distributed energy balance model to a mid-latitude glacier, using a combination of previously published parameter values and values estimated from local point-scale measurements, and found reasonable agreement for local energy fluxes, but poor results for the glacier-
wide mass balance. The albedo parametrization was identified as a potential source of error as it was based on data from a single point (Klok and Oerlemans, 2002; Oerlemans and Knapp, 1998) and may not be valid elsewhere on the glacier surface throughout all seasons (Van De Wal et al., 1992; Konzelmann and Braithwaite, 1995).

In studies of spatially distributed glacier mass balance (e.g. Klok and Oerlemans, 2004; Hock and Holmgren, 2005; Hock, 2005; Reijmer and Hock, 2007; Mölg et al., 2009; Rye et al., 2012; Gurgiser et al., 2013) optimization of free parameters to *in*
*situ* measurements can be successful if the processes being parametrized are quasi-constant over the whole glacier surface, or if a dense measurement network is available for spatially-distributed optimization. Brock et al. (2000) concludes that the accuracy of spatially distributed models is strongly dependent on the ability to apply multiple local optimizations, and on the importance of individual energy components. Nevertheless, most of the temperature index models (Hock, 2005; Pellicciotti et al., 2005;





Carenzo et al., 2009; Robinson et al., 2010, 2011) and also a number of energy balance models (Mölg et al., 2009; Gurgiser et al., 2013) have been optimized towards a single best fit to the glacier-wide mass balance measurement, which requires a subjective choice of the single mass balance metric to be used. For example optimizing for cumulative mass balance, mass balance gradient or stake measurements have been shown to be problematic as different optimal solutions are found depending on the mass balance metric chosen for optimization (Rye et al., 2012). The associated differences in the individual optimal parameter values and resultant values of the energy components have not been studied in detail, and furthermore, published uncertainties of mass balance measurements (Zemp et al., 2013; Galos et al., 2017) imply that a single best fit model simulation may not be found at all (Beven and Binley, 1992).

A more powerful way forward may be found in multi-objective optimization of glacier energy balance modeling, first applied by Rye et al. (2012). They optimized a mass and energy balance model, on two Arctic glaciers in Svalbard over ~40 years using three objectives for optimization: (i) the mass balance gradient, (ii) the mean absolute error (MAE) at the stake location, and (iii) the cumulative mass balance. This approach creates an ensemble of optimal solutions which all are equally 'good' in respect to all three objectives. With this approach they could reconstruct the mass balance of the glaciers before direct measurements were available and also give an estimate of the model uncertainty from the parameter spread within the optimal solution set. This work demonstrated that it is likely that stated model performance based on single objective optimizations do not adequately represent model performance at a glacier scale or over longer time periods.

Mass balance models are required to be transferable in space and time in order to estimate run-off on a larger scale or the impact of a changing climate (Oerlemans et al., 2005; De Woul and Hock, 2005; Raper and Braithwaite, 2006). Studies of transferability of an enhanced temperature-index model (Carenzo et al., 2009) used the optimized parameters from one particular year and glacier and compared it to the locally optimized run at different glaciers and over different time periods. They concluded that their model shows a rather good transferability in space, except during overcast conditions. Furthermore, they observe that the parameters vary depending on year and location and are correlated to each other. MacDougall and Flowers (2010) and Prinz et al. (2016) investigate transferability of full energy balance models: While MacDougall and Flowers (2010) find satisfactory temporal transferability in the Arctic over two years, albeit with some local parameter adjustment, Prinz et al. (2016) fails to do so in the tropics over an interval of a century. This is attributed to a substantially changed climate over the century and/or a different micro-meteorological setting due to dramatic glacier shrinkage (Prinz et al., 2016). This implies the problem of transferring a calibrated model to rather different climatic settings/glaciers and raises the question about the general uncertainty and transferability of such models.

It can be expected that models with more parameters have greater variation in the solutions. Reduction of free parameters for optimization based on a sensitivity analysis is therefore a helpful tool to reduce both the effect of parameter correlation and computational expense (Spear and Hornberger, 1980; Saltelli et al., 2000; van Griensven et al., 2006). For example Gurgiser et al. (2013) applied such a parameter reduction procedure on a tropical glacier to reduce the free parameters prior to assessing model transferability.

Many previous studies do not separate model sensitivity and model uncertainty in a transparent manner. Hence, model uncertainty assessments of varying robustness have been presented in the literature. For example, Mölg et al. (2012) used a





simplified approach to quantify uncertainty of the mass balance model used in this study: An arbitrarily chosen spread of the most positive and negative deviation simulations around their single best fit in respect to Root Mean Square Deviation (RMSD) of cumulative mass balance is used to estimate uncertainty. This gives only a very rough estimate as only two particular runs determine the uncertainty estimate. Anslow et al. (2008) first optimize their model and then vary the optimized parameters

within certain bounds (5 %) and perturb the meteorological input to quantify the impact on the mass balance. This provides the sensitivity of the model output towards the parameter values and inputs, but the created range is also used as model error estimate. Such approaches are inadequate as (i) they lack a global uncertainty estimate, (ii) *a priori* setting of the parameter optimum is needed to assess the sensitivity, and (iii) the model uncertainty is limited by allowing only a small range in parameter variation. Machguth et al. (2008) perform a similar assessment but base their perturbation ranges on probability density

functions whereby model uncertainty is assessed by applying random and systematic errors/uncertainties to the meteorological input data as well as to the mean value of parameters. Considering such uncertainties in meteorological input offers an opportunity to quantify the resulting uncertainty in the final model output, but a direct model uncertainty quantification based on the model structure/parametrizations is not revealed, and applying random and systematic errors to arbitrarily chosen parameters is poorly constrained. The reported uncertainty, of $700\,\mathrm{kg\,m^{-2}}$ for 400 days at a single point (roughly 10 % of the total melt),

is related to the standard deviation of the probability density function.

    In this study we target a clear separation of the concepts of sensitivity and uncertainty in an assessment of the performance of a distributed mass and energy balance model (Mölg and Hardy, 2004; Mölg et al., 2008; Mölg et al., 2009) using three years of summer mass balances simulated on two mid-latitude glaciers. This is achieved by first applying a global sensitivity analysis to reduce the parameter space. This is an extension of the local sensitivity analysis used by Gurgiser et al. (2013) to

a global variance based method (Saltelli et al., 2006), which has recently been applied in snow pack modeling (Sauter and Obleitner, 2015). Subsequently we build upon the multi-objective optimization applied by Rye et al. (2012) to quantify the model output, the parameter uncertainty and resulting uncertainty of the energy components based on a set of three objective functions used for Monte Carlo model optimization. The temporal and spatial transfer of such a model ensemble is assessed with cross-validation. Finally, the uncertainty of the model in the resulting energy components is presented. The aim is to

develop a workflow for a more rigorous assessment of model performance that can quantify the uncertainty of the modeling chain applied.

## 2   Study sites and model input data

Two glaciers in the Eastern European Alps were selected as test sites in this study (Fig. 1). Hintereisferner (HEF; 46.80°N, 10.75°E) is a sizeable valley glacier in the Austrian Ötztal-Alps spanning 3720 to 2454 m a.s.l. in 2013, when the glacier area

was ca. 6.7 km² and Langenferner/Vedretta Lunga (LGF; 46.46°N, 10.61°E) is a smaller valley glacier in the Italian Ortler-Alps spanning 3370 to 2711 m a.s.l. in 2013. These glaciers were chosen since the model used here requires topographic and meteorological input data, and measurements of surface mass balance for evaluation. For both these glaciers (i) topographic data is available in the form of high-resolution digital elevation models (DEMs) derived from airborne laser-scanning data



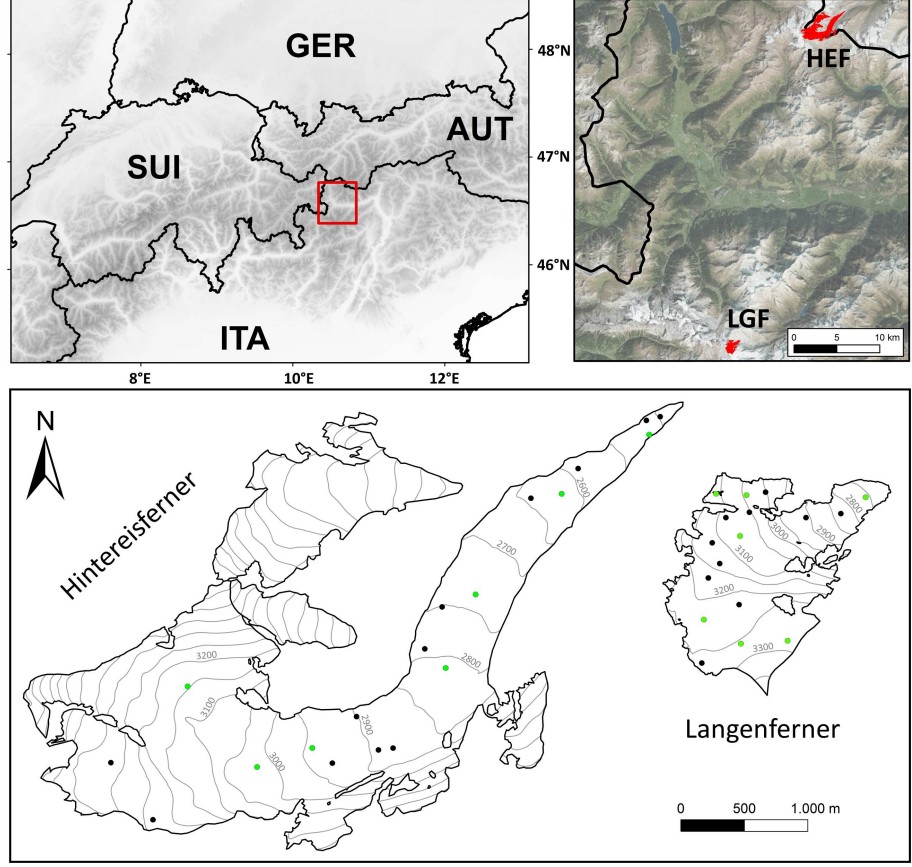

**Figure 1.** Model simulations are performed at the stake locations shown as points; points marked in black are only used in the optimization, while green points indicate the seven stakes on each glacier that were also used in the sensitivity analysis. Detailed maps are available in the supplement (fig. S.1-2).

acquired in Fall 2013 (Galos et al., 2015); (ii) meteorological data are available from automatic weather stations (AWSs) in the vicinity of the glaciers for the period 2012 to 2014 and (iii) intense glaciological observations, including measurements of seasonal mass balance (e.g. Klug et al., 2017; Galos et al., 2017), are available.

At HEF the AWS is located on a small plateau within a rock slope north of the upper tongue area of the glacier at an altitude of 3025 m a.s.l.. The horizontal distance of this AWS to the glacier is about 300 m and it provides all meteorological data required for the model except for precipitation. Precipitation data was taken from the gauge operated by the Bavarian Academy of Sciences at Vernagtbrücke, 3.5 km east of HEF at an elevation of 2600 m a.s.l., and scaled to the elevation of the AWS on the basis of precipitation gradients derived from 11 totalizing rain gauges in the vicinity of the glacier (Strasser et al., 2017). At LGF the AWS data come from the station of the Hydrological Service of the province of Bozen operated at Sulden Madritsch, 2.5 km north of the glacier at an altitude of 2825 m a.s.l. (Galos et al., 2017).



## 3 Model and methods

### 3.1 Energy balance model

The energy and mass balance model used in this study is a process-based model that has been applied in a range of glacier environments (Mölg and Hardy, 2004; Mölg et al., 2008; Mölg et al., 2009, 2012; Gurgiser et al., 2013; Prinz et al., 2016; Galos et al., 2017). The model was run with hourly time-steps for three summer periods over each glacier. The model is a distributed mass and energy balance model, but in this study simulations were limited to 18 stake locations on each glacier to reduce computational expense. The model tracks the accumulation of solid precipitation and uses the surface energy balance to calculate the ablation at the glacier surface:

$$Q_M + Q_{ice} = SW_{net} + LW_{net} + Q_S + Q_L + Q_G + Q_P \tag{1}$$

where $LW_{net}$, $SW_{net}$ are the net radiation balances for long-wave (thermal) and short-wave (solar) radiation and the other energy fluxes are the sensible ($Q_S$), latent ($Q_L$), ground ($Q_G$) and precipitation ($Q_P$) heat flux. The available energy is used to raise the glacier surface temperature ($Q_{ice}$) if below freezing point or for melting ($Q_M$) if the glacier surface is at the melting point. Mass losses of the glacier are represented via melt ($Q_M$) and sublimation ($Q_L$). We use the model in a similar configuration to Prinz et al., 2016. The only difference is given by a change in the shortwave radiation scheme which is explained in the detailed model description in Appendix A1-A6.

### 3.2 Methods

#### 3.2.1 Global Sensitivity Analysis (GSA)

Variance based sensitivity testing methods work in a probabilistic framework judging sensitivity by relative variances of model input and output (van Griensven et al., 2006; Saltelli et al., 2000, 2006, 2010). This is a global method that is independent of model calibration i.e. independent of a local optimal run, and is hereafter referred to as Global Sensitivity Analysis (GSA). The method treats the model as a simple function $f$ with:

$$y = f(X) \ \ X = X_1, X_2, ..., X_n \tag{2}$$

where $y$ is the single model result (in this case mass balance) and $X_{1,...,n}$ are the individual input parameters.

The influence of an individual parameter can be examined by the main effect ($V_i$) of $X_i$ on $Y$.

$$V_i = V_{Xi}(E_{X-i}(Y|X_i)) \tag{3}$$

$X_{-i}$ is the whole parameter space except any variation in $X_i$ (a fixed $X_i$), $E$ is the expectation value and $V$ the variance. $E_{X-i}(Y|X_i)$ is the mean model output with whole parameter variation except in $X_i$. The variance over all values for $X_i$ yield the variance attributed to parameter $X_i$. The sensitivity of the model towards single parameters is evaluated by normalizing by the total variance of the output.

$$S_{Xi} = \frac{V_{Xi}(E_{X-i}(Y|X_i))}{V_y} \tag{4}$$



$S_{Xi}$ is the first order sensitivity index. The total sensitivity index $(S_{Ti})$ is the effect of $X_i$ with all its interactions on the model variance:

$$S_{Ti} = \frac{E_{X-i}(V_{Xi}(Y|X_{-i}))}{V_y} \tag{5}$$

This can be related to the sensitivity obtained from local sensitivity analysis. The model sensitivity (variance) to $X_i$ is tested $(V_{Xi}(Y|X_{-i}))$ at every point of the parameter space ($X-i$ fixed). To clarify, consider the example of a simple non-additive model $Y = X_1 \cdot X_2 + X_3$ with the variables $X_i$ as input parameters with a given variance/uncertainty. Assuming unified distribution within the intervals

$$X_1 \in [1,3], X_2 \in [0.1,0.3], X_3 \in [0.5,1]$$

leads to a model output range of $Y \in [0.6, 1.9]$. The variance-based method yield the results for $S_{Xi}$, the first order sensitivity
index and $S_{Ti}$, the total sensitivity index for an ensemble of 10,000 runs as shown in Table 1. The first order effect of $X_3$ is the largest, while the other two are similar if computational uncertainty is neglected. Most variance is caused by the last parameter. $X_3$ has no interactions, so its total index is the same as the first order one, while interaction between $X_1$ and $X_2$ creates additional variance, so their total index is higher. In the example $X_1$ and $X_2$ contribute to $\approx$ 60 % of the total variance and $X_3 \approx$ 40 %, as $X_1 \cdot X_2 \in [0.1, 0.9]$ and $X_3 \in [0.5, 1]$.

The estimation of the sensitivity indices follows the algorithm from Saltelli et al. (2010). The model used here has 23 free parameters. A base sample of 12,000 parameter settings was created with a quasi-random Sobol sequence. The random numbers are linearly transformed onto the parameter intervals. The distribution is always treated as uniform and the limits for every parameter are given in Table (2). The indices are estimated with $N \cdot (k+2)$ runs, where $k$ is the number of parameters and $N$ the base sample size. The GSA consisted of a total ensemble size of 300,000 simulations per year and glacier, fulfilling the
convergence criteria for the algorithm. To reduce computational expenses the GSA model was limited to seven stake locations on each glacier (Fig. 1).

   The parameter sensitivity results from the GSA are also used as a tool to reduce the number of free parameters in the model by identifying those parameters which have only a marginal influence on the model output (Spear and Hornberger, 1980; Saltelli et al., 2000; van Griensven et al., 2006). The model is considered insensitive to parameters with a total sensitivity
index $(S_{Ti})$ of <0.05, and these parameters were fixed at the median value of the range shown in Table 2 in subsequent model simulations.

**Table 1.** The sensitivity indices for the simple model $Y = X_1 \cdot X_2 + X_3$. The indices for $X_1$ and $X_2$ are similar as they both have the same normalized variance. $X_1 \cdot X_2$ creates additional variance by the interaction of the two parameters yielding higher total indices.

|          | $X_1$ | $X_2$ | $X_3$ |
|----------|-------|-------|-------|
| $S_{Xi}$ | 0.26  | 0.27  | 0.43  |
| $S_{Ti}$ | 0.31  | 0.30  | 0.43  |





**Table 2.** In the sensitivity analysis 23 different parameters were used. The range used in the sensitivity study for each parameter is given here. The equations of most of the parametrizations are given in Appendix A.

| # | Name | Abbreviation | minimum | maximum | unit |
|---|------|-------------|---------|---------|------|
| 1 | temperature gradient | $T_{grad}$ | 0.0055 | 0.0085 | Km$^{-1}$ |
| 2 | precipitation gradient | $P_{grad}$ | 0 | 0.12 | m$^{-1}$ |
| 3 | all liquid precipitation threshold | $P_{limit+}$ | 2 | 3 | °C |
| 4 | all solid precipitation threshold | $P_{limit-}$ | 0.5 | 1.5 | °C |
| 5 | surface layer thickness | $sfc$ | 0.1 | 0.5 | m |
| 6 | momentum roughness length (ice) | $z_{0i}$ | $1 \cdot 10^{-3}$ | $5 \cdot 10^{-3}$ | m |
| 7 | scalar roughness length over ice | $z_{hi}$ | $0.1 \cdot 10^{-3}$ | $2 \cdot 10^{-3}$ | m |
| 8 | roughness length over fresh snow | $z_{hfs}$ | $0.1 \cdot 10^{-3}$ | $2 \cdot 10^{-3}$ | m |
| 9 | momentum roughness length over old snow | $z_{0fs}$ | $1.5 \cdot 10^{-3}$ | $6.5 \cdot 10^{-3}$ | m |
| 10 | precipitation density | $\rho_s$ | 200 | 370 | kgm$^{-3}$ |
| 11 | part of refreezing mass forming superimposed ice | $suifra$ | 0.0 | 0.36 | |
| 12 | absorbed shortwave at ice surface | $\zeta_i$ | 0.72 | 0.88 | |
| 13 | absorbed shortwave at snow surface | $\zeta_s$ | 0.81 | 0.99 | |
| 14 | extinction coefficient of ice | $\beta_i$ | 2 | 3 | |
| 15 | extinction coefficient of snow | $\beta_s$ | 13.68 | 20.52 | |
| 16 | value for bottom temperature | $T_{bottom}$ | 271 | 273 | K |
| 17 | ice-albedo | $\alpha_i$ | 0.15 | 0.25 | |
| 18 | fresh-snow-albedo | $\alpha_{fs}$ | 0.8 | 0.9 | |
| 19 | firn-albedo | $\alpha_{fi}$ | 0.4 | 0.65 | |
| 20 | timescale in albedo module | $t$ | 5 | 30 | days |
| 21 | depth-scale in albedo module | $d$ | 2 | 5 | cm |
| (22) | precipitation perturbation | $P_{pertu}$ | -10 | +10 | % |
| 23 | roughness length of aged snow | $z_{0hfi}$ | $0.1 \cdot 10^{-3}$ | $4 \cdot 10^{-3}$ | m |

### 3.2.2 Multi-objective optimization and uncertainty quantification

A multi-objective optimization allows for more than one optimal solution in the calibration procedure, and offers a potential quantification of model uncertainties. The multi-objective optimization used here follows previous approaches in hydrology and glaciology (Yapo et al., 1998; Rye et al., 2012). Where the model is given $n$ objectives, with $f_n$ to be minimized in respect to the model parameter input $X$, the optimization approach can be written as:

$$minimize(f_1(X), f_1(X), ..., f_n(X)) \tag{6}$$





The result of Eq. (6) is an ensemble of optimal solutions that represent trade-offs between the objectives and no single one can be deemed superior to the other optimal solutions. Therefore, they are called the non-dominated set of optimal solutions, or Pareto set (Pareto, 1971). As an illustration, consider an optimization with two objectives $(f_1, f_2)$: The concept of a Pareto optimal set is shown in Fig. 2 in which the (classic) single objective solutions are the points $f_1^{min}$ and $f_2^{min}$ for the two

5   objectives respectively. A solution at the utopian point is desirable as all functions would be at their minimum, but the models generally cannot optimize the different objectives simultaneously. There are only compromise solutions between the objectives. The members of the set of optimal solutions defining the Pareto front are superior to the other solutions, but are all equal to each other without subjective ranking by the modeler. The variation of the parameters of the optimal solution set defines the minimum parameter uncertainty (Vrugt et al., 2007). This uncertainty is a result of shortcomings in the model and/or the

10  variations of parameters, such as spatial or temporal change in true parameter value over the simulation period (Oerlemans and Greuell, 1986; Marshall and Warren, 1987). If a single simulation must be chosen to be the optimal model set up, the compromise solution, defined as the point with the lowest euclidean distance to the utopian point is a common choice.

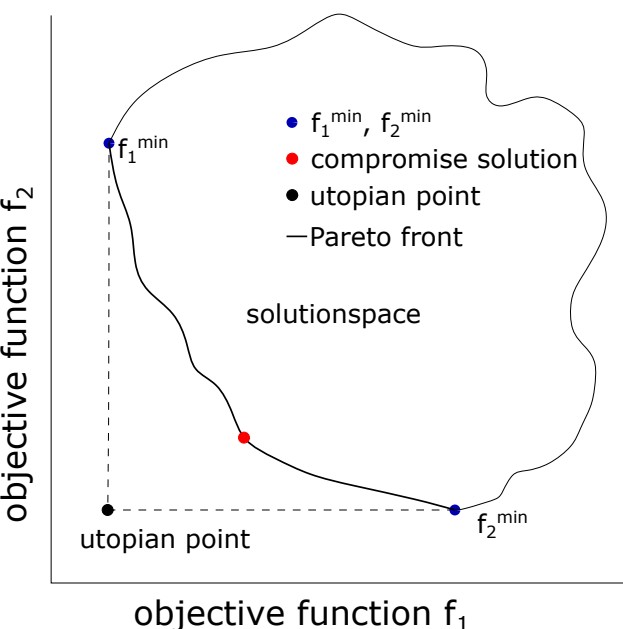

**Figure 2.** The figure displays a two-dimensional Pareto-space which comprises a 2-dimensional Pareto front. The solutions on this front (black solid line) are referred to as the non-dominated set of solutions. In comparison all other solutions within the solution space are inferior in at least one objective relatively to the Pareto front. Classic single objective optimization yields the points $f_1^{min}$ and $f_2^{min}$, which represent the minimum of those objectives that the model can achieve. The utopian point (black) is the point $(f_1^{min}, f_2^{min})$ where both objectives are at their minimal value. Commonly the compromise solution (red) of the Pareto-set is considered an objective choice for a single solution as it has the minimum euclidean distance of the optimal solution towards the utopian point.





In this study the multi-objective optimization is based on a Monte Carlo simulation. The non-sensitive parameters from the GSA were fixed to their median value from the range used in the GSA (Table 2). Then 20,000 model simulations with random parameter combinations of the remaining parameters were created and the mass and energy balance simulated for 18 stake locations. This approach was chosen above an evolutionary algorithm so that different objective function spaces and all single

objectives can be investigated with the same set of simulations. Various objective functions were initially explored including Root Mean Square Deviation and Mean Absolute Deviation over all simulation points, but finally three objective functions that captured the main patterns of behaviour were applied: (i) the $BIAS$ over all simulated stakes, (ii) the mean absolute deviation ($MAD$) of the lower 9 stakes ($MAD_{low9}$) and (iii) the $MAD$ over the upper 9 stakes ($MAD_{top9}$). The $BIAS$ is used as a proxy for the cumulative mass balance with avoiding of interpolation errors. The RMSD is a commonly used measure

for optimization in glaciological modeling (Gurgiser et al., 2013). By using the $MAD$ here we want to reduce the effect of individual stakes which could be influenced by processes which are not governed by the model, but the general feature of those two statistical functions are similar. Previous studies (e.q. Klok and Oerlemans, 2004; Hock, 2005; Sauter and Obleitner, 2015) have focussed on the accumulation and ablation area separately or exclusively, but without a distinct mathematical comparison. Therefore the approach of the split $MAD$ was chosen. The Pareto front was identified, and additionally a second ensemble

including solutions within a certain range (500 kgm$^{-2}$) from the Pareto front, was identified to account for errors in the field measurements of mass balance at each stake simulation point. Results of this second ensemble will only be mentioned briefly throughout the discussion. The spread of the parameter settings of all optimal solutions of the Pareto and near-Pareto sets are used to indicate the parameter uncertainty for each case, and the calculated surface energy balance components of these optimal sets are also used to estimate the uncertainty of the energy components on the point scale, as well as on the glacier scale.

## 4   Results and Discussion

### 4.1   Global sensitivity analysis

The focus of this GSA is not on the absolute sensitivity towards single parameters, but rather to reduce the dimension of the parameter space. Therefore, the following discussion is limited to two classes: parameters to which the model is sensitive ($S_{Ti} > 0.05$) and non-sensitive ($S_{Ti} < 0.05$). On each glacier the mass and energy balance at 7 stake locations over three years

was simulated for the GSA, so the maximum count of sensitivity for a parameter would be 21, meaning that the model is always sensitive to that parameter at every point of the glacier.

At Hintereisferner, 11 out of 23 parameters are identified as sensitive (Fig. 3 (a)), and these sensitive parameters are classified in two general categories. Firstly, all but the lowest stake location are sensitive to parameters related to surface albedo, particularly of snow and firn, and secondly, for stakes with high elevation differences compared to the AWS, the model is also

sensitive to the vertical temperature gradient.

The sensitivities show spatial and temporal variability which can be explained by the varying mass balance conditions of the respective year (mean specific summer/annual mass balance with 2012 -2643/ − 1560, 2013 −1841/ − 510 and 2014 −1494/ − 122 kgm$^{-2}$). For example, sensitivity towards the ice-related parameters is most evident in 2012, which was the

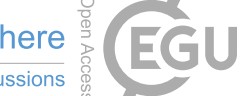

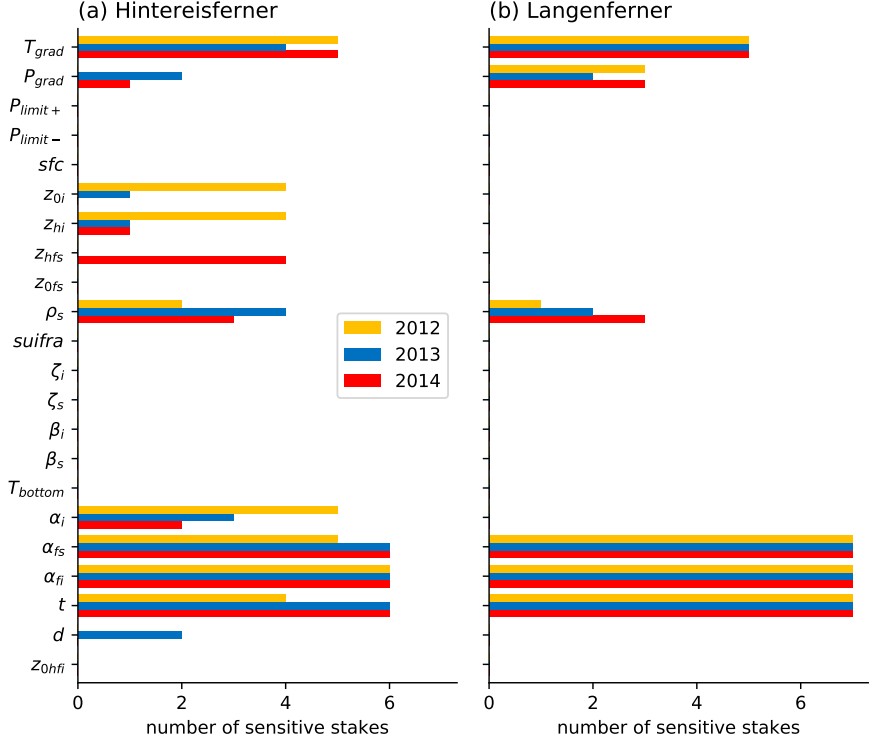

**Figure 3.** The amount of sensitive stakes per year for (a) HEF and (b) LGF. The sensitivity analysis was performed at 7 stakes on each glacier, though the vertical gradients can only be tested at 6 stakes as one is located at the same altitude as the reference weather station. Every parameter with a sensitivity index higher than 0.05 gets a score of 1, giving a maximum count of 7 per year (meaning the model is sensitive to this parameter at all stakes). Parameters involved in the parametrization of surface albedo are dominating, with snow related values in the upper section of the glacier and ice at the lower stakes. Hintereisferner shows a total of 11 sensitive parameters and Langenferner 6.

driest and most negative mass balance year, with large parts of the glacier surface free of snow and firn for most of the ablation season. The roughness length of fresh snow, by contrast is only influential at the upper stakes in 2014, where snow fall was frequent during the ablation season resulting in the least negative mass balance of the three study years. Sensitivity towards the elevational precipitation gradient is only relevant at the lowest stakes (500 m below the weather station) in the wet years.

5      On the smaller Langenferner 6 of the 23 parameters were identified as sensitive (Fig. 3 (b)). Similar as at HEF, the model shows consistent sensitivity to surface albedo and the vertical temperature gradient. As LGF is smaller than HEF, the sensitivity shows less variability in space and time, though the annual mass balances during the three study years range from -1500 to +400 kgm$^{-2}$, and, as the tongue of LGF does not extend to such a low elevation as the one of HEF, it is less sensitive to ice-related parameters. Variations in the ice albedo within the bounds of 0.15 and 0.25 hardly influence the mass balance model results on

10     the smaller glacier, even though ice is exposed for the majority of the summer at the lowest stake. This low sensitivity to the





**Table 3.** Five objective functions are used to analyze the model performance. The minimum value for every function each year are given in kgm$^{-2}$. While the $BIAS$ is low in all cases, absolute errors and $RMSD$ are much higher, and highest in 2012. The minimum $MAD$ is not the same run over the whole glacier and its upper/lower parts.

|  | HEF 2012 | HEF 2013 | HEF 2014 | LGF 2012 | LGF 2013 | LGF 2014 |
|---|---|---|---|---|---|---|
| $BIAS$ | 0,11 | 0,48 | 0,00 | 0,52 | 0,28 | 0,04 |
| $RMSD$ | 470 | 213 | 285 | 537 | 391 | 214 |
| $MAD$ | 414 | 170 | 225 | 419 | 309 | 153 |
| $MAD_{top9}$ | 252 | 108 | 228 | 328 | 114 | 170 |
| $MAD_{low9}$ | 397 | 165 | 130 | 346 | 283 | 81 |

ice albedo compared to the snow albedo parameters is explained by the fact that, as the removal of snow cover is accompanied by a large drop in albedo (0.4-0.65 to 0.15-0.25), the time of exposure is more crucial than the final ice albedo, and this time of ice exposure is itself influenced by the snow albedo via its dominant control on the short-wave radiation budget. Within the chosen parameter ranges, the net short-wave radiation varies by 50 % in case of fresh snow (10-20 % absorbed) and only by 12 % over ice.

## 4.2 Calibration

First we consider the best model performance with respect to each individual objective function tested (Table 3), before presenting the multi-objective optimization based on the first, fourth and fifth objective in Fig. 4.

In all cases a model simulation with very low bias (<1 kgm$^{-2}$) with respect to the stake mass balance can be found. This illustrates that apparently good optimization on the single value of cumulative mass balance over the stakes is relatively easy to achieve (Table 3). In comparison, the deviation in all other objective functions is much higher, ranging from 81-537 kgm$^{-2}$. The deviations in these objectives are all largest in 2012 on both glaciers. $RMSD$ and $MAD$ vary similarly between the years at each glacier, with the higher $RMSD$ values indicating a non-uniform deviation from the measurements over the stakes. With the exception of 2014, the glacier-averaged $MAD$ is larger than the $MAD$ calculated for either the upper/lower section of the glaciers. This is to be expected as the stakes within each section of the glacier experience more similar climate conditions, resulting in a lower $MAD$. The fact that $MAD$ in the lower glacier section is larger than in the upper section in 2012 and 2013 is probably related to the incapability of the model to correctly reproduce the date of ice exposure. In 2014 the upper glacier sections show a slightly higher $MAD$, associated with above average accumulation in the previous winter and the frequent summer snowfall in this season.

The multi-objective optimization, using $BIAS$, $MAD_{top9}$ and $MAD_{low9}$, yields an ensemble of solutions. The non-dominated set for each of the three years has 27, 17, 69 members for HEF and 58, 61, 14 members for LGF respectively (fig. S.4). The fewest solutions are found in years with the lowest total $MAD$ (HEF 2013, LGF 2014). Figure 4 shows the Pareto-front of optimal solutions for HEF 2012 and the corresponding parameter settings. A low bias is easily achieved by the



model if no other objectives are considered because it is a single value (the sum of the mass balance at all stakes) and, for example, deviations in the ablation and accumulation area may cancel each other. The projections onto the $BIAS$ planes are less curved (the distance between the utopian and compromise point is lower) and the performance in respect to the $MADs$ can be drastically increased with only a small cost in the $BIAS$. The two-dimensional projections of the Pareto-space (Fig. 4 (a) and (b)) illustrates, for example, that allowing for a model bias of 25 kgm$^{-2}$ can improve the $MAD$ by 200 and 300 kgm$^{-2}$ in the lower and upper glacier sections respectively. The $MADs$ plane (Fig. 4 (c)) is more curved, indicating that the two objectives cannot be optimized by the model at the same time, such that some parameter sets leading to good results for the ablation zone of the glacier may not sufficiently reproduce the relevant processes in the accumulation zone.

The parameter values of those optimal solutions span the entire allowed space apart for some of those relating to snow albedo which span (almost) the whole parameter space in all years for both glaciers, and show no obvious tendencies towards a certain albedo range (S. 3). For HEF in 2012, snow albedo values cluster in the higher range (0.52-0.6) for firn and (0.86-0.9) for fresh snow (Fig. 4 (d)), while on LGF lower firn and fresh snow albedo values (<0.5/<0.84) are optimal. Similar behavior is observed for the albedo time scale (see Appendix A3) which tends towards higher values for HEF in 2012/13 and towards lower ones for LGF in 2013/14. The confinement of snow albedo is mainly a result of the highest model sensitivity towards this parameter, nevertheless it still varies and the converse argument, of less sensitive parameters showing greater span is not valid: For example, the roughness length over fresh snow is generally at the lower margin of the allowed parameter range (0.1-0.14·10$^{-3}$) in 2012 even though the model is considered insensitive ($S_{Tz_{hfs}} < 0.05$) to this parameter in the particular year. These results highlight that the parameter settings of multiple optimal solutions for this type of mass and energy balance model can vary drastically. There are no clear correlations between two individual parameters, instead all parameters interact simultaneously to some degree. Without the *a priori* reduction of model parameters by GSA even less information could be extracted from the optimization. Compared to Rye et al. (2012) our results appear less constrained which can be explained by the narrow initial parameter ranges used in our study. Rye et al. (2012) for example applied values for fresh snow albedo in the range of 0.65 to 0.95, while we restricted the initial range to values between 0.8 and 0.9 as reported in the literature (e.q. Cuffey and Paterson, 2010).

## 4.3 Transferability studies

To investigate the transferability of the optimized mass balance model settings, all the optimal solutions of the Pareto set of one glacier summer mass balance case were applied to the five other summer and glacier cases. While, each Pareto solution set was identified based on the multi-objective optimization, the transferability study uses only the euclidean distance towards the utopian point as a quantification tool. In Fig. 5 the optimal solutions for HEF 2013 and their performance in the other model periods is shown. The individual optimal parameter settings yield quite different mass-balance-values/performances for the other summers. While the performance on the same glacier (HEF) is reasonably good for 2012 (200-800 kgm$^{-2}$ compared to 152-600 kgm$^{-2}$ in the optimization period of summer 2013) and slightly worse for 2014, the optimal solutions do not perform so well for LGF, resulting in euclidean distances of up to 3500 kgm$^{-2}$. Analogous analysis of the ensemble behavior of other summers shows that the optimal solution for 2012 also performs well in 2013 and vice-versa, and show acceptable performance



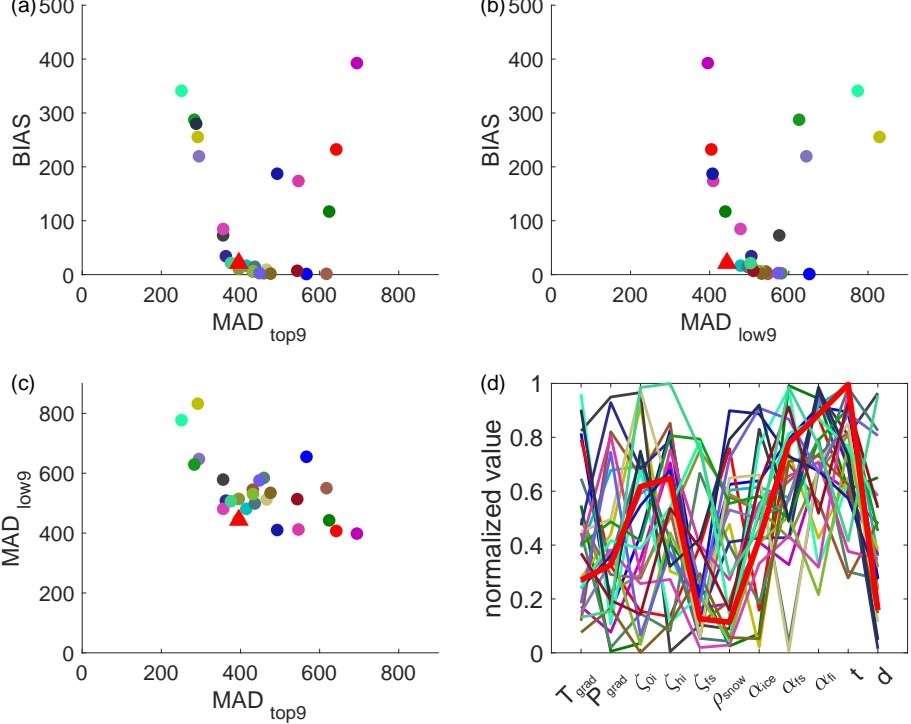

**Figure 4.** Each individual member of the Pareto set for HEF in 2012 is displayed with a different color and the compromise solution highlighted (red triangle/red line). The different panels are the two-dimensional projections of the Pareto-space onto the (a) $BIAS$ and $MAD_{top9}$; (b) $BIAS$ and $MAD_{low9}$; (c) $MAD_{low9}$ and $MAD_{top9}$ planes. (d) Shows the normalized parameter values for each case in the same colors as in the Pareto space plots with all parameters apart firn albedo and albedo timescale spanning over the entire parameter space. It furthermore shows that a single solution is not representative for the ensemble.

in 2014 respectively. The deviation of the 2012 and 2013 optimal values of HEF yield errors greater than 2000 kgm$^{-2}$ on LGF. The 2014 HEF ensemble performs on average better on HEF, but two simulations perform better on LGF in 2012/13 and around 20 are within the same error as for HEF. On LGF also 2012 and 2013 agree better, and do produce reasonable results for both glaciers in 2014. The ensemble of 2014 on LGF yields similar errors (250-800 kgm$^{-2}$) for LGF 12/13 and HEF 14.

5   All ensembles of LGF produce larger errors on HEF in 2012 and 2013.

The cross validation (Fig. 6) focuses on the transferability of the single compromise solution to other season and glacier cases. This can be considered as a classical best guess solution. The features follow the structure of the ensemble behavior discussed above with HEF 2012 and 2013 seeming to be distinct from the other four cases. The compromise solutions for HEF 2012 and 2013 are similar in performance and parameter value and, while they perform adequately for HEF in 2014, within

10   the estimated uncertainty of 1300 kgm$^{-2}$, the error is greater than 1500 kgm$^{-2}$ when either of these compromise solutions is applied on LGF, no matter for which year. Similarly, the compromise solution for the three year period for HEF ($RMSD_{HEF}$ in Fig. 6), which is dominated by the characteristics of 2012 and 2013, also performs poorly when applied to LGF (errors




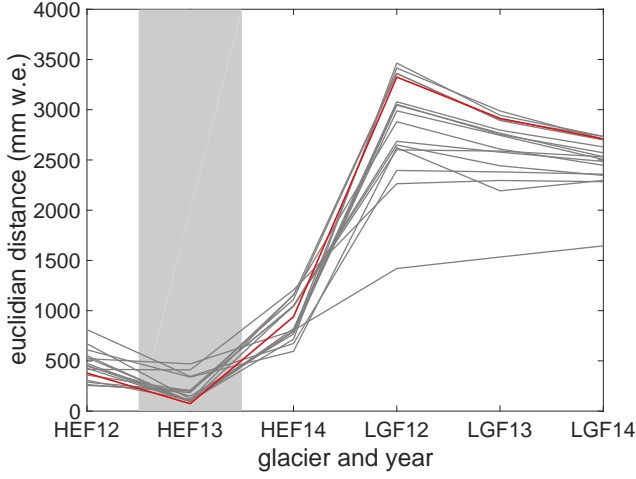

**Figure 5.** The euclidean distance of the 17 optimal solutions for model parametrization that comprise the Pareto set for HEF in 2013 as applied to all six glacier/summer cases. The performance in 2012 on HEF is still reasonably good and slightly worse for 2014. The optimal solutions for HEF 2013 perform worse in all years on LGF than in any years at HEF.

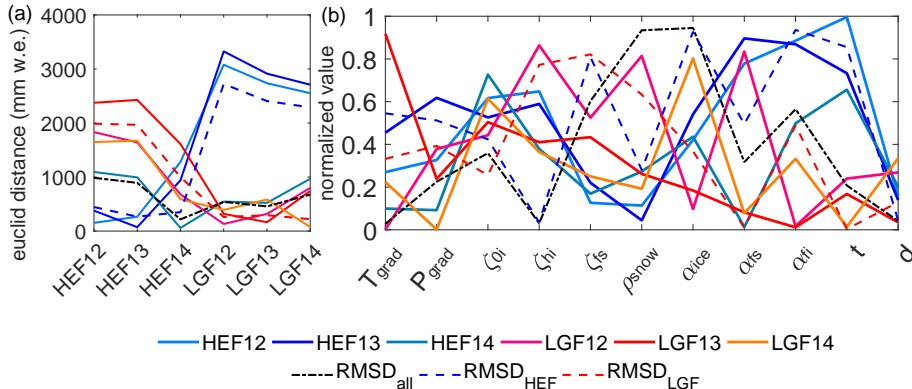

**Figure 6.** (a) Performance of single compromise solution for each season and glacier ($GGG_{yy}$), with HEF in solid blue colors and LGF in red. The simulations which perform best over a three ($RMSD_{HEF}$ and $RMSD_{LGF}$), and six year period ($RMSD_{all}$) respectively are given with dashed lines, following the same color scheme. (b) The corresponding parameter setting of the optimal solutions to the left. The color scheme is equivalent. The snow-albedo related parameters take rather large values on HEF and small on LGF. No clear trend is visibile for the other parameters.





of up to 3500 kgm$^{-2}$). The compromise solution of HEF 2014, however, generally performs better on LGF than for other years at HEF, and reciprocally, the compromise solution over the whole period at LGF performs best at HEF in 2014, and the maximum error (up to 2500 kgm$^{-2}$) is lower than for cases of HEF compromise solutions being applied to LGF. This is probably due to the domination of more negative mass balances in 2012 and 2013 at HEF, where good model performance is

linked to capturing the large extent of the ablation area, whereas the shorter glacier tongue at LGF has smaller impact on the mass balance of this glacier. The compromise solution ($RMSD_{all}$) for all six cases also highlights that within this set of six the cases HEF 2012 and HEF 2013 are more distinct from the other cases as the overall compromise solution performs worst in these two cases. For most parameters no clear separation between the two glaciers is evident, except for fresh snow albedo and the albedo timescale which are both larger at HEF and smaller at LGF. Inspection of the optimal parameter values reveals

that runs with a longer calibration period ($RMSD_{xxx}$) do not necessarily take trade-off values between the individual years. For example, in this case the solution that performs best over both glaciers and the whole time period ($RMSD_{all}$) takes larger values of fresh snow density and ice-albedo than any other compromise solution (Fig. 6(b)). This further highlights the model complexity and is suggestive of the effects of physical shortcomings (such as parameter values that are constant in space and time) cancelling each other out.

## 4.4   Energy balance components

Analysis of the energy balance components associated with Pareto set solutions offers a qualitative means of verifying that the identified optimal parameter settings are inkeeping with expected physical processes at the glacier surface. The energy balance components calculated by the model are expected to vary depending on the parameter settings of an optimal ensemble, which have been demonstrated to span almost the whole parameter space. This variation in energy balance components is indicative

of the uncertainty in the modeled energy fluxes. Figure 7 illustrates such variations in the energy balance components for the case of HEF 2012. In this case, the most uncertain energy balance component is the short-wave radiation, which at the same time is the largest energy source for the surface. Total energy flux from short wave radiation decreases with altitude, while the associated uncertainty increases. The sensible and latent heat flux provide a net energy source to the surface and their value and uncertainty also decrease with altitude. The long-wave radiation budget is a net energy loss from the surface in summer

and its value increases, and its uncertainty decreases, with elevation. As a result of these elevational patterns in uncertainty, the uncertainty in energy melt energy is also largest at low elevations.

The variation of the averaged energy components over the stakes for HEF 2012 are given in Fig. 8. The uncertainties are generally lower than on a stake basis. The short-wave, conductive ground heat flux and sensible heat flux supply a net heating to the surface on both glaciers. The precipitation heat flux is also a minor energy source. The penetration of short-wave radiation

and the long-wave budget remove energy from the glacier surface. Latent heat is the only energy flux that has either a positive or negative effect on the surface energy balance depending on stake location, glacier and year. On both glaciers lower elevation locations tend to have more positive energy fluxes from latent heat. At HEF this flux is mostly an energy addition to the glacier surface while on LGF it mostly serves to remove energy from the surface. In the beginning of the summer, sublimation during the day and condensation/re-sublimation during the night is dominant on HEF, and the general trend over the summer is to



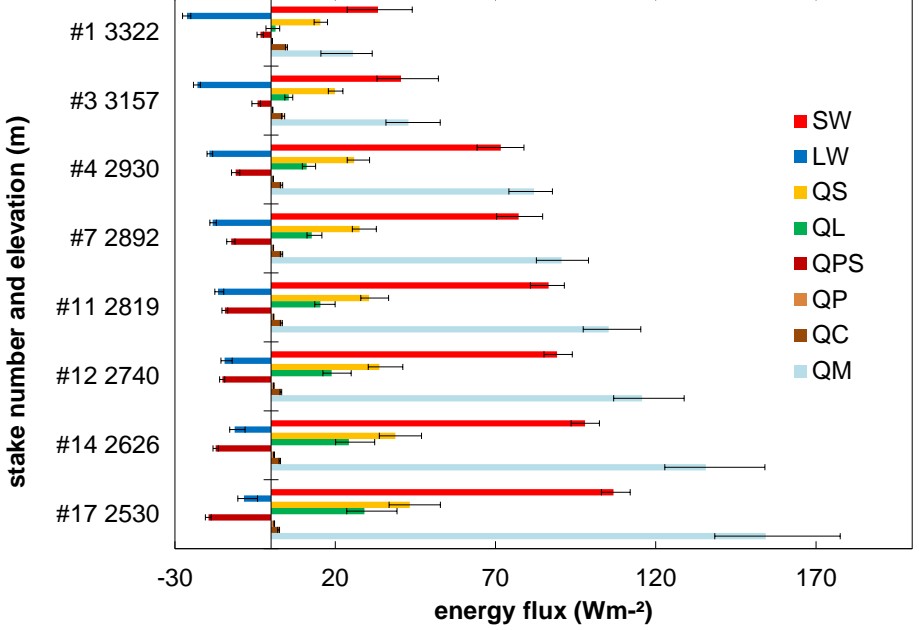

**Figure 7.** The energy balance components for 8/18 selected stake locations close to the central flow line, displayed in different colors for HEF 2012. Solid bars represent the fluxes of short-wave radiation ($SW$), long-wave radiation ($LW$), turbulent heat fluxes ($Q_L$ and $Q_S$), penetrating short-wave ($Q_{PS}$), precipitation heat flux ($Q_P$), conductive heat flux ($Q_C$) and the resultant available heat for melting ($Q_M$, here plotted as a positive flux).

progressively more condensation. LGF shows less condensation during (the) mid-summer, which is mainly attributed to less windy conditions than at HEF.

The total contribution of the energy balance components averaged over the glacier are listed in Table 4. The relative uncertainties of the energy balance components are up to 50 % of their contribution on single stake basis and 30 % averaged over

HEF; slightly lower (30 and 25 % respectively) for LGF. This leads to a variation in the available heat for melting and the mass balance of about 30 % on a point scale. The absolute uncertainty of the seasonally averaged available energy for melting can reach up to 35 Wm$^{-2}$ at the tongue area of HEF. This corresponds to a daily melt uncertainty of 9 kgm$^{-2}$ and seasonal uncertainty of up to 1.3 m w.e.. The glacier averaged available heat for melting is much less uncertain over all stakes. This is a result of the calibration process. The sum of total available melt energy is directly linked to the bias as objective function,

which shows the largest value among the optimal solutions on HEF 2012 with 600 kgm$^{-2}$. In comparison the $MADs$ which are more influenced by the mass balance at the individual stake reach values up to 1000 kgm$^{-2}$.

The largest uncertainty of this energy balance model is associated with the short wave radiation as a result of the albedo parametrization, which relies on five model parameters. Alternative albedo parametrizations are also known to be a source of substantial uncertainty in other studies (Willeit and Ganopolski, 2017). The greatest uncertainty is found in the accumulation

area and around the equilibrium line altitude. This is because (i) the parametrization for snow albedo has more variation/free



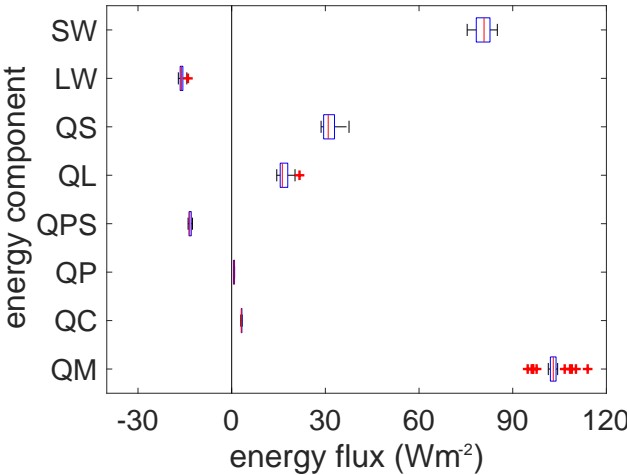

**Figure 8.** The energy balance components average over all stakes has less uncertainty than on the point scale for HEF 2012. As the objective functions are all integrated over the whole glacier and therefore the uncertainty is lower. Glacier wide the short-wave radiation is the largest component with also the largest absolute uncertainty, followed by the turbulent fluxes. The long-wave balance and the penetrating short-wave radiation provide a net cooling effect for the surface.

**Table 4.** The energy balance components are averaged over all stake locations. The uncertainty is given in respect to the minimum and maximum of the ensemble. The short-wave radiation ($SW_{net}$) has the largest impact, decreasing in importance from 2012 to 14, with a less negative mass balance ($Q_M$). The penetrating shortwave radiation ($Q_{PS}$) follows the same pattern with opposite effect. The long-wave budget ($LW_{net}$) is lower for LGF. The turbulent fluxes are greatest in 2012 and larger on HEF. The precipitation ($Q_P$) and convective ($Q_C$) heat flux are of minor importance.

| | $SW_{net}$ | $LW_{net}$ | $Q_S$ | $Q_L$ | $Q_{PS}$ | $Q_P$ | $Q_M$ | $Q_C$ | |
|---|---|---|---|---|---|---|---|---|---|
| HEF 12 | $80 \pm 10$ | $-16 \pm 3$ | $31 \pm 9$ | $17 \pm 7$ | $-13 \pm 1$ | 1 | $103 \pm 19$ | 3 | Wm$^{-2}$ |
| HEF 13 | $75 \pm 11$ | $-21 \pm 3$ | $21 \pm 7$ | $13 \pm 5$ | $-12 \pm 2$ | 0 | $80 \pm 8$ | 4 | Wm$^{-2}$ |
| HEF 14 | $69 \pm 15$ | $-21 \pm 3$ | $20 \pm 7$ | $8 \pm 5$ | $-10 \pm 2$ | 1 | $71 \pm 7$ | 4 | Wm$^{-2}$ |
| LGF 12 | $122 \pm 14$ | $-22 \pm 1$ | $14 \pm 4$ | $-3 \pm 1$ | $-19 \pm 3$ | 1 | $97 \pm 11$ | 4 | Wm$^{-2}$ |
| LGF 13 | $112 \pm 22$ | $-28 \pm 2$ | $8 \pm 3$ | $-3 \pm 2$ | $-16 \pm 4$ | 0 | $78 \pm 16$ | $5 \pm 1$ | Wm$^{-2}$ |
| LGF 14 | $95 \pm 7$ | $-27 \pm 1$ | $9 \pm 2$ | $-2 \pm 1$ | $-12 \pm 1$ | 1 | $68 \pm 5$ | 5 | Wm$^{-2}$ |



model parameters than albedo over ice and (ii) around the ELA the variation of the ice exposure date increases the uncertainty of short-wave radiation flux. Point scale albedo measurements combined with localized optimization schemes may solve this issue, but for distributed models a more detailed model may be necessary to better capture the full complexity of the processes governing initial snow albedo and its chnge through time (Flanner and Zender, 2006).

The long-wave radiation has a lower uncertainty in this study than in Sauter and Obleitner (2015) and its uncertainty is mainly due to the air temperature, the related temperature gradient parameter, and the surface temperature. We cannot state that the general uncertainty of energy balance models associated with incoming long-wave radiation is low, rather the parametrization was optimized prior to the sensitivity analysis as direct measurements are available at the weather station. The parametrization gives no bias for the station but hourly RMSD was up to 30 Wm$^{-2}$, which is in the range of the net long-wave budget.

This therefore also mainly influences short term differences in the long-wave budget rather than the seasonal energy flux. Nevertheless, as with albedo, it remains unclear whether long-wave radiation modules based on air-temperature, cloudiness and sky-view factor are sufficient to model spatio-temporal variation over a glacier.

The turbulent fluxes are associated with the second largest uncertainty, which is in agreement with other studies finding larger uncertainties in the radiative forcing (Willis et al., 2002). Turbulent fluxes are important for determining short-term

variations of melt rates due to, for example, changes in the stability regimes (Lang, 1981). The uncertainties in our model are due to differences in roughness lengths and the temperature gradient. Roughness lengths over ice and snow vary substantially (Braithwaite, 1995, e.g.) in space and time (Greuell and Konzelmann, 1994; Calanca, 2001), and also with wind speed. The appropriateness of using constants for these values in glacier modelling is also questionable, and stability corrections may differ from the glacier margins to the interior, for example. It is therefore also questionable how appropriate constant roughness

lengths and stability corrections for ice and snow in space and time are.

The energy balance model used here indicates that it is important to treat penetrating short wave radiation in the surface energy balance, though its effects are difficult to confirm by empirical measurements. In agreement with Hock (2005) we can conclude that heat supplied by rain is negligible in the mid-latitudes.

## 4.5   Implications of this study

The larger glacier, Hintereisferner, has more sensitive parameters and the variation over the stakes is larger than at Langenferner, as a result of more distinct climate regions on the longer tongue of the larger glacier. This is also true for the uncertainty of energy balance components, with the exception of the net solar radiation, which is comparable on both glaciers. Short-wave radiation is the most uncertain of the energy balance components, due to the albedo parametrization, which accounts for the change in albedo over time, but does not account for any possible spatial variation in temperature or grainsize-dependent albedo

decay rates. We have shown that the model has difficulties to optimize the upper and lower part of the glacier simultaneously, as a result of the variable values of physical quantities like albedo. The large spread of our ensemble is a result of trade-off solutions between the real albedo at any time at any location and the temporally and spatially averaged parametrization applied. Other parameterizations that are assumed constant in space and/or time, or only indirectly varied by temperature and altitude dependencies, are also subject to similar trade off effects. Although the physical relations may not be the same at all times and at




the lower tongue area may be quite different from the upper glacier, this does not mean that model performance is worse on the larger glacier (HEF) with more variation in a quantitative matter (Table. 3), but rather that the solutions of the Pareto front show more variation in the parameter settings. This analysis clearly identifies the issue of governing parameters/parameterizations not being constant in space and time as the main problem of distributed energy balance modeling; the most readily appreciable

example of which is ice albedo which is often lower nearer the terminus due to debris and dust accumulation and water saturation of the glacier surface.

We see two potential approaches to improve this: (1) Although for a broad range of applications, optimizing all key parameters serves a purpose, fixing low sensitivity parameters to common values, which are not optimized, results in a type of a simplification of the model that reduces over-fitting and potentially increases the stability and comparability of the energy bal-

ance model over short-timescales. The overall performance of such a model will be lower because the tuning possibilities have been restricted, but better estimates of the uncertainties for out-of-sample periods can be generated. (2) Parameters or parameterizations could be allowed to vary in space and/or time. This could be achieved either by increasing the measurements/data availability or increasing the model complexity. For example snow albedo as well as surface roughness length depend on the grain size, which in turn could be based on melt rates in the model and lapsed time since the last snowfall. Parametrizing this

requires more field data to constrain the physical process and should not be just added as additional model free parameters to optimize.

The approaches in this study are helpful tools to combine these suggestions. A clear understanding of the model sensitivity, independent of the optimization of the model is necessary to decide on the importance of certain parameters. It gives the option to fix parameters and focus on the key processes. We have shown that the multi-objective optimization is a valid tool to asses

uncertainties in the model. The objectives used are all based on the same data (i.e. stake data). This allowed us to show the uncertainty that is just associated with treating the available data in a different way without requiring additional measurements. The model can readily be optimized to minimise bias or meet any single value objective, therefore model performance based on single best fit approaches should be treated with caution. The mass balance as an objective should always be considered with $RMSD$ or $MAD$ too. The chosen objectives show there is inter-annual variation in the performances of the upper and

lower section of the glacier in our cases. The curved nature of the Pareto front highlights that simultaneous optimization of both areas is difficult for the model. Parameters are just not constant, in either space or time, so the uncertainty increases when the model is applied to other time periods or on another glacier. The overall uncertainty is in the range of 1 kgm$^{-2}$. It is larger when transferring the calibrated model to another alpine glacier, but still of the same order of magnitude. Together with an uncertainty estimation of the energy balance components the key parametrizations, which need further improvement, can be

identified. Within the multi-objective framework it is furthermore possible to focus on processes individually: For example if the albedo is measured on the point scale, the difference to its model value could be used as an objective, instead of *a priori* calibration of the albedo parametrization itself.

Neither meteorological forcing on the point scale nor mass balance measurements are absolute, and these uncertainties were not formally included in this study. Zemp et al. (2013) have estimated an annual measurement uncertainty of 140 kgm$^{-2}$ on

point scale glaciological mass balance measurements. More information about the propagation of this error would need to be




known to quantitatively include it in the optimization. But if 50 kgm$^{-2}$ uncertainty in the $MAD$ and $BIAS$ is included, the Pareto sets increase by one order of magnitude making interpretations harder and further increases the total output uncertainty.

The analysis presented here indicate that while mass and energy balance models help us to understand the physical processes on the glacier, the necessity for parameterizations within these models introduces considerable, variable uncertainty to the model output. Calibration of surface mass balance models is complex and uncertainty studies are helpful to understand the models, and it is not advisable to draw substantial conclusions from such modeling efforts without first fully understanding the inherent model sensitivity and the properties of the uncertainty of the calculated mass balance and associated energy fluxes in detail.

## 5 Conclusions

Based on a well developed mass and energy balance model, applied to two well-studied glaciers in the European Alps, this study gives a robust estimate of the model uncertainty and discusses the advantages of parameter space reduction and multi-objective optimization in glaciological modeling.

Using a variance based global sensitivity method model sensitivity to the model free parameters was identified, independent of the calibration data. Model sensitivity to specific parameters is both site- and time- specific, and this should be acknowledged in wider applications of such models. By separating the parameters into two sensitivity categories the model parameters to be optimized can be reduced. Those that the model output is sensitive to were subject to a multi-objective optimization, while non-sensitive parameters were fixed to literature values.

The multi-objective optimization was based on three objectives related to stake mass balance data measured using the glaciological method. We used the model bias over all stakes and the mean absolute deviation over the upper and lower part of the glaciers. It proved difficult to optimize model performance in the upper and lower section of the glacier simultaneously. The bias over all stakes, which was used as a proxy for the cumulative mass balance, can be minimized easily, and this should be considered when optimizing for a single best fit against single values. The ensemble of optimal solutions shows a wide spread of parameter settings within the physically reasonable range. This implies that the common approach of a single best optimized parameter set is subject to over-fitting and may significantly differ from other equally plausible solutions. Furthermore, our results show that the constraint of plausible parameters is only marginally linked to the sensitivity, with very sensitive parameters also taking multiple optimal values. This implies that keeping these parameters constant in space and time increases the uncertainty. The overall model uncertainty is in the range of 1 kgm$^{-2}$ over the whole ensemble, and increases when applied to the other glacier and years. The model performance is worse when applied to another glacier, but is of the same order of magnitude as the temporal transfer, suggesting the model can be applied, within its uncertainty, to other glaciers with similar climatic settings.

Parameter uncertainty is connected with uncertainty in the energy balance components, which, in the cases studied here, reached 30 % averaged over the glacier and 50 % at individual measurement stake locations. TFor the model used here, the most uncertain energy balance components are net short-wave radiation and turbulent fluxes. Reasserting the findings of other





studies that indicate the snow and ice albedo representation is the most crucial parameter on mid-latitudes glaciers for the summer mass balance.

Overall the findings of this study highlight that understanding the sensitivity and uncertainty of surface energy and mass balance models is complex, and simplistic assessments of model performance are likely to overstate the model capabilities.

Further studies such as this, incorporating more models, glaciers and years would help constrain the degree to which results from such models can be considered reliable for regional applications and for projections of glacier mass balance.

*Code availability.* The code of the mass balance model can be requested from Thomas Mölg (thomas.moelg@fau.de). Pareto construction scripts and the updated solar module can furthermore be requested directly from Tobias Zolles (tobias.zolles@uib.no).

*Data availability.* The used mass balance and meteorological data is available at zenodo.org; DOI:10.5281/zenodo.1326398. All mass bal-
ance data is publicly available through the WGMS (https://wgms.ch/).

## Appendix A: Model description

The mass and energy balance model used here consists of coupled surface and subsurface components. The model computes mass balance as the sum of solid precipitation, surface deposition, internal accumulation (refreezing of liquid water in snow), change in englacial liquid water storage, subsurface and surface melt, and sublimation. This approach is based on the surface

energy balance of a glacier in the following form:

$$Q_M + Q_{ice} = SW_{net} + LW_{net} + Q_S + Q_L + Q_G + Q_P \tag{A1}$$

where $SW_{net}$ is net short-wave radiation, $LW_{net}$ is the sum if incoming and outgoing long-wave radiation a the glacier surface, $Q_S$ and $Q_L$ are the turbulent fluxes of sensible and latent heat, respectively, $Q_G$ is the subsurface energy flux comprised of $Q_C$, the conductive heat flux in the subsurface, and $Q_{PS}$ the energy flux from short-wave radiation penetrating into the subsurface,

and finally, $Q_P$ is the heat flux from precipitation. The sum of these fluxes yields a residual flux F which, if the glacier surface temperature (TS) reaches 273.15 K, represents the latent energy for melting. If TS is below 273.15 K, energy conservation is achieved by solving TS to balance the fluxes (e.q. Mölg et al., 2009). The model is fully described in the previously mentioned publications and briefly below.

### A1 Long-wave radiation

The calculation of the incoming long-wave radiation is based on Stefan-Boltzmann law (Mölg et al., 2009; Klok and Oerlemans, 2002; Konzelmann et al., 1994):

$$LW_{in} = \sigma \epsilon T_a.^4 \tag{A2}$$



with $\sigma$ being the Stefan-Boltzmann constant and $\epsilon$ the emissivity:

$$\epsilon = \epsilon_{cs}(1 - n^p) + \epsilon_{cl}n^p \tag{A3}$$

where $cs$ and $cl$ are the clear-sky and cloud emissivity respectively, $n$ is the cloud cover fraction calculated in the solar module as $n_{eff}$ and $p$ an exponent related to the importance of cloud emissivity (Greuell et al., 1997). The cloud emissivity is computed using

$$\epsilon_{cl} = 0.23 + b(\frac{e_a}{T_a})^{1/8} \tag{A4}$$

with $e_a$ as the atmospheric vapor pressure. The three parameters $\epsilon_{cs}$, $p$ and $b$ were optimized (using a 5000 member Monte Carlo) to reproduce the measured long-wave radiation. First the runs within 10 % of the best run in respect to a weighted average of BIAS and RMSD between the simulated and the measured incoming long-wave radiation at the HEF Station were determined. The run of this ensemble with the lowest RMSD/BIAS on LGF was taken as the best compromise solution. The parameters are fixed within the model for the whole study period and are based on three summers of data at HEF and 1.5 at LGF (therefore a larger impact of the longer data at HEF on the optimization). The trade-off values are taken to be applicable on both glaciers with the final values of $b = 0.515$, $n = 1.95$ and $\epsilon_{cs} = 0.994$. These setting results in an hourly RMSD below 31/37 Wm$^{-2}$ for HEF/LGF and no bias, this is not far of the optimal setting for either glacier with 30/36 Wm$^{-2}$.

The outgoing long-wave radiation follows Stefan-Boltzmann law Eq. (A2), with $T$ the glacier surface temperature and the emissivity of ice $\epsilon_i$ is assumed 1.

## A2 Convective fluxes

The latent heat flux ($Q_L$) and the sensible ($Q_S$) are computed similar to Mölg and Hardy (2004). The calculations are based on Monin-Obhukov similarity theory (Garratt et al., 1992).

$$Q_L = 0.623 L_v \rho_0 \frac{1}{p_0} \frac{\kappa^2 \nu(e_a - E_s)}{ln\frac{z_m}{z_{0m}} ln\frac{z_\nu}{z_{0\nu}}} \tag{A5}$$

with $L_v$ being the enthalpy of vaporization ($2.514 MJkg^{-1}$), $\rho_0$ the air density at mean sea level (1.29 kgm$^{-3}$), $p_0$ is $1013 hPa$, $\kappa$ the van Karman constant (0.4), $e_a$ is the water vapor pressure in air and $E_s$ the surface value respectively. $z_{0m}$ and $z_{0\nu}$ are the momentum and scalar roughness length of water vapor. $z_m$ and $z_v$ is the height above ground where the wind speed and the water vapor ($e_a$) is measured/calculated. The sensible heat flux

$$Q_S = c_p \rho_0 \frac{p}{p_0} \frac{\kappa^2 \nu(T_a - T_s)}{ln\frac{z_m}{z_{0m}} ln\frac{z_h}{z_{0h}}} \tag{A6}$$



is computed with $c_p$ the specific heat of air at constant pressure, $T_a$, $T_S$ the air and surface temperature and $z_h$ the scalar roughness length for temperature. The roughness length $(z_j)$ are model free parameters in this study. The model distinguishes three different roughness lengths depending on the glacier surface: fresh snow, firn and ice. For a stable stratified atmosphere a stability correction based on Phi functions is applied (Mölg and Hardy, 2004).

**A3    Surface albedo and the Albedo-module**

The albedo parametrization is based on Oerlemans and Knapp (1998). It computes the broad band albedo for each grid cell, based on the ice and snow albedo and the depth of the snow pack:

$$\alpha = \alpha_{snow} + (\alpha_{ice} - \alpha_{snow}) \cdot \exp(\frac{-d}{d^*}) \tag{A7}$$

$\alpha_{ice}$ is a model free parameter, $d$ is the snow depth, and $d^*$ is the characteristic scale for the snow depth and a free parameter (Oerlemans and Knapp, 1998). The relation for the snow albedo $(\alpha_{snow})$ is

$$\alpha_{snow} = \alpha_{firn} + (\alpha_{freshsnow} - \alpha_{firn}) \cdot \exp(\frac{-t}{t^*}) \tag{A8}$$

with $\alpha_{firn}$, $\alpha_{freshsnow}$ and $t^*$ as model free parameters subject of/to optimization. The albedo module $(t^*)$ is a characteristic time scale in days (Klok and Oerlemans, 2002) and $t$ the time since the last snowfall event ($> 0.1cm$ fresh snow).

**A4    Surface Temperature and ground energy flux**

The conductive heat flux $(Q_C)$ and the energy flux from penetrating shortwave radiation $(Q_{PS})$ determine the ground heat flux $(Q_G)$ of the energy balance (EQ. (1)). The model solves the thermodynamic energy equation for a multi-layer grid with a fixed bottom temperature (15 Layers, 0.1m steps in the first meter, gradually increasing to a total depth of 7 m). The bottom temperature is a model free parameter. $Q_C$ is computed from the temperature difference between the surface and the first layer.

The calculation of the penetration of short-wave radiation is based on Bintanja and Van Den Broeke (1995). A constant fraction $(1 - \zeta_i)$ of the net-shortwave radiation is penetrating the surface and the intensity is exponentially decreasing with depth. The optimization and sensitivity analysis in this study uses four parameters with the extinction coefficient and the absorbed fraction $(\zeta_i)$ for snow and ice.

**A5    Surface accumulation/precipitation**

The surface accumulation is directly related to the precipitation. The model has two threshold values for all liquid and all solid precipitation (Mölg et al., 2012). In between these the portion increases linearly. The temperature threshold as well as the density of solid precipitation are subject of the sensitivity analysis and optimization.





## A6 Solar module and solar module sensitivity

The parametrization of the short-wave radiation is based on the calculation of the cloudiness, in the form of the effective cloud cover fraction $n_eff$:

$$n_{eff} = \frac{1 - SW_{mea}/(D_{cs} + S_{cs})}{k} \tag{A9}$$

with $SW_{mea}$ being the measured short-wave and $D_{cs}, S_{cs}$ the calculated diffuse and direct radiation under clear sky conditions. The parameter $k$ determines at which fraction of the clear sky value full cloudiness is achieved i.e. all incoming radiation is diffuse. (Important to note, we allow $n_{eff} > 1$ if such low radiation was measured.) The influence of $k$ on the model output was investigated (Appendix A6). The calculation of the clear sky values is described in Mölg et al. (2009). The diffusion portion of radiation under clear sky conditions was determined using a manual selection of clear sky days. The values var-

ied between the snow free ($K_{dif} = 0.51$) and snow covered days ($K_{dif} > 0.65$). For the calculations an averaged value of 0.6 was used. As $K_{dif}$ is a fixed glacier wide value, while snow cover might vary, a modulation depending on the conditions at the weather station is not possible. The applicability of $K_{dif}$ as a single value might need to be reevaluated for other models/applications/research questions.

The calculation of the incoming short-wave radiation on every point of the glacier is based on the assumption of homoge-

neous cloudiness ($n_{eff}$). It is a reversing of Eq. (A9):

$$SW_{diff} = (D_{cs} + S_{cs}) \cdot (1 - n_{eff} \cdot k)((1 - p_{diff}) \cdot n_{eff} + p_{diff}) \tag{A10}$$

with $SW_{diff}$ being the calculated diffuse radiation and $p_{diff}$ the portion of diffuse radiation under clear sky conditions. $p_{diff}$ is calculated as the ratio of the clear sky diffuse and total radiation. It was 0.084 and 0.085 for the two glaciers and set to 8.5 % (for both to have a common value). Compared to previous works using the solar module, we changed the increase of diffuse

radiation. Instead of a linear increase of diffuse radiation, the portion of diffuse radiation is linearly increasing with increased cloudiness. This is a basic parametrization and reproduces the measured radiation fully. Via $n_{eff}$ $k$ is determining the ratio of direct and diffuse radiation. This could alter the energy balance. The direct radiation is calculated analogous and corrected for slope and aspect.

The calculation of solar radiation incorporates the free parameter $k$, which determines at which fraction of the total possible

global radiation everything is considered as diffuse radiation. The parameter $k$ varies with latitude (Hastenrath, 1984) an is not constant in time either, therefore the effective cloud cover incorporates some of its variability and is not exactly the cloudiness (Mölg et al., 2009). With the new used parametrization (eq. A10) the global solar radiation at the weather station can be fully reproduced so $k$ cannot be optimized. But it determines the portions of direct and diffuse radiation, which may have a significant influence on the energy and mass balance. Therefore, an additional GSA was performed with the parameter $k$ as the 24th model

free parameter. Based on the values for the tropics 0.65 (Mölg et al., 2009) and the arctic with $\approx 0.85$ (Hock and Holmgren, 2005) it was varied in this range for the sensitivity analysis. Its maximum sensitivity index over all 7 investigated stakes in the GSA was $2 \times 10^{-3}$, which is one order of magnitude lower than the threshold for our sensitive parameters. Therefore, the choice of $k$ within the given range is not influential on the simulation of the mass balance on/at the glacier. The model albedo



does not vary between direct and diffuse radiation, so it only influences the total amount of radiation at less/more shaded areas than the weather station.

Furthermore, the change in the calculation of direct and diffuse components from linear with cloudiness to a linear increase of the fraction are better suited to represent the site radiation. This is in agreement with measured radiation by Hock and

Holmgren (2005) on the Arctic glacier, Storglaciären (fig. S. 5). The slightly higher starting value ($p_{diff}$) is due to larger portion of diffuse radiation under clear sky conditions in the arctic than in the mid-latitudes and a higher final value is due to a smaller $k$ in this study with 0.8 compared to around 0.85 in the arctic. The influence of this change in parametrization is probably also rather small, as the model is not sensitive to changes in the relative fractions of diffuse and direct radiation on the chosen glaciers/stake location.

*Author contributions.*  TZ conducted the simulations and the data analysis and wrote the main part of the manuscript. FM was involved in defining the study and contributed to the statistical analysis. WG was involved in the model set-up and the adaption of the solar module. SG was responsible for the mass balance measurements and data acquisition. LN contributed to the paper design and writing. All authors contributed to finalizing the manuscript.

*Competing interests.*  The authors declare that they have no conflict of interest.

*Acknowledgements.*  The computational results presented have been achieved (in part) using the HPC infrastructure LEO of the University of Innsbruck. We thank Rainer Prinz for supplying the snow, density and DEM grids for the energy balance model. We thank the Hydrographisches Amt Bozen and the Hydrographischer Dienst Tirol for funding the mass balance measurements. LN was supported by the Austrian Science Fund Grant number V309.





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
