# Peer review of "Robust uncertainty assessment of the spatio-temporal transferability of glacier mass and energy balance models"

_The Cryosphere, 2018_

## Referee Comment (RC1) · M. Lafaysse (Referee) · 8 Oct 2018

The manuscript of Zolles et al presents ensemble simulations of glacier mass balance with an energy balance model applied on 2 glaciers and 3 seasons. The main innovation compared to the existing literature consists in a relatively comprehensive analysis of model sensitivities and uncertainties. The paper is well written and well structured. The statistical framework is clearly explained. I especially appreciate the effort of the authors to give simple examples to explicit the formalism (pages 7; 9). The conclusions are clearly summarized and consistent with the obtained results. The complex equifinality between the parameterizations of such models is well demonstrated and the

implication in model calibration and model transferability is very interesting. Therefore, I think this paper deserves publication after a minor revision which would account for my few comments below when it is possible.

Page 2 line 19: I think most studies base this statement on an evaluation of the energy fluxes and surface temperature, not only the melt rates.

Page 4 line 7: I understand the deficiencies of the cited references but given the number of studies which just present simulation outputs without any uncertainty quantification, I think that the word "inadequate" is a bit severe.

Page 4 line 11 / Table 2: the 23 parameters include a "precipitation perturbation" which disappears in the results section (Figure 3) without any specific explanation.

More generally, it is not completely clear if the authors want to incorporate the spatialization of meteorological data as part of their model uncertainty study. The decision to exclude the longwave parameterization from the free parameters has a strong consequence in the results. Indeed, large errors are introduced here because Equation A2 is a strong simplification of the real behaviour of the full column of atmosphere. Snow models are usually extremely sensitive to these errors (Sauter and Obleitner, 2015; Quéno et al, 2017). The authors acknowledge this limitation (page 19 lines 5-12) but I do not really understand this choice. Why should the impact of temperature gradient uncertainty on longwave radiation be accounted for if the parameters of equations A3 and A4 are not? Similarly, what is the logic in considering the uncertainty of precipitation gradient but not the uncertainty of the mean precipitation forcing? I think it could also be more explicit in the text that Figure 7 does not represent the full contributions of uncertainties. The very narrow range obtained for longwave radiation is unlikely to represent the real uncertainty of this component as the incoming flux is highly uncertain whereas it is not accounted for.

Table 2: Can you comment the range of precipitation density? This range is not realistic for snowfall in the Alpine area (too high values, Helfricht et al, 2018). It may

compensate some deficiency in a simple model which does not represent accurately compaction but this should be detailed. The authors could also comment the implication in the uncertainty analysis of using some potentially unrealistic values for some parameter ranges. The same could apply if the precipitation perturbation was really considered because a 10% error is not sufficient to represent precipitation uncertainty in moutainous areas.

Page 13 lines 21-22: I am not sure to correctly understand this sentence. Could you develop what you mean by "less constrained" and what is the relationship with a narrow initial range of parameters?

Page 19 line 1-4: This is true but rather utopic at the moment. Such models need a forcing of impurity depositions. The existing products are not sufficiently reliable nor sufficiently detailed to depict the processes responsible for the spatial variability of albedo on a glacier.

Page 19 lines 13-20: The authors discuss the impact of the possible variability of roughness lengths. However, I think they could also discuss more generally the relevance of applying this theory of turbulent fluxes formulation in mountainous environments where the turbulence is probably more affected by the surrounding topography than by the surface roughness itself (Conway and Cullen, 2013).

Page 19 lines 21-22 Which effects are you talking about? From experiments with a detailed snowpack model (with a sufficient vertical discretization), it is rather clear than the absorption profile has an impact on surface temperature and on the temperature gradient close to the surface (and therefore on snow metamorphism). However, the effect on more integrated variables is likely to be much less significant.

Page 20 line 15 I do not know whether new field experiments on that topic are really required right now. The authors should first mention that the relationship between albedo and grain shapes and sizes is already implemented in detailed snowpack models such as Crocus (Vionnet et al, 2012) or SNOWPACK (Lehning et al, 2002).

[Figure]

Page 20 lines 30-32 I agree and the same applies for various variables, especially surface temperature which is a good indicator of the correct resolution of the energy balance.

Page 20 line 33 The lack of a full quantification of the meteorological uncertainty is probably the main limitation of this paper. This is only stated here in a small paragraph which would have deserved to be more developed based on the existing literature. Indeed, this is probably the most studied uncertainty in previous studies in snow modelling (e.g. Raleigh et al, 2015) and in glacier modelling. However, the possible compensation errors between meteorological forcing and model parameters may deteriorate the relevance of model uncertainty studies which do not account for forcing uncertainties. I did the same thing myself in the context of a detailed multiphysics snowpack modelling (Lafaysse et al, 2017) but I just think that this limitation could be more discussed.

Page 21 line 27 Âă: To what does 1 kg/m$^2$ refer? In which duration?

Typos: Abstract line 5: "which" introduces Page 16 line 26: energy melt energy Page 19 line 4: change Page 21 line 32: For

References

Conway, J. and Cullen, N.: Constraining turbulent heat flux parameterization over a temperate maritime glacier in New Zealand, Annals of Glaciol., 54, 41-51, doi:10.3189/2013AoG63A604, 2013

Helfricht, K., Hartl, L., Koch, R., Marty, C., Olefs, M.: Obtaining sub-daily new snow density from automated measurements in high mountain regions, Hydrol. Earth Syst. Sci., 22, 2655-2668, doi:10.5194/hess-22-2655-2018, 2018

Lafaysse, M., Cluzet, B., Dumont, M., Lejeune, Y., Vionnet, V. and Morin, S.: A multiphysical ensemble system of numerical snow modelling, The Cryosphere, 11, 1173-1198, doi:10.5194/tc-11-1173-2017, 2017

Lehning, M., Bartelt, P., Brown, B., Fierz, C., and Satyawali, P.: A physical SNOWPACK model for the Swiss avalanche warning, Part II: snow microstructure, Cold Reg. Sci. Technol., 35, 147– 167, doi:10.1016/S0165-232X(02)00073-3, 2002

Raleigh, M. S., Lundquist, J. D., and Clark, M. P.: Exploring the impact of forcing error characteristics on physically based snow simulations within a global sensitivity analysis framework, Hydrol. Earth Syst. Sci., 19, 3153–3179, doi:10.5194/hess-19-3153- 2015, 2015.

Quéno L., Karbou F., Vionnet V., Dombrowski-Etchevers I. : Satellite products of incoming solar and longwave radiations used for snowpack modelling in mountainous terrain, Hydrol. Earth Syst. Sci. Discuss., doi:10.5194/hess-2017-563" in review.

Sauter, T. and Obleitner, F.: Assessing the uncertainty of glacier mass-balance simulations in the European Arctic based on variance decomposition, Geosci. Model Dev., 8, 3911–3928, doi:10.5194/gmd-8-3911-2015, 2015.

Vionnet, V., Brun, E., Morin, S., Boone, A., Faroux, S., Le Moigne, P., Martin, E., and Willemet, J.-M.: The detailed snowpack scheme Crocus and its implementation in SURFEX v7.2, Geosci. Model Dev., 5, 773–791, doi:10.5194/gmd-5-773-2012, 2012.

---

## Short Comment (SC1) · 26 Oct 2018

**1. Review**

The paper proposed a two-step procedure to assess the uncertainties in a widely used distributed glacier mass and energy balance models. First the global sensitivity analysis identifies the sensitivity of the parameters. The reduction of parameters through sensitivity test and the optimization of model by allowing a wider range in parameter variation yields promising results. Then the best setting parameters are determined by a Monte Carlo multi-objective optimization. The transferability is investigated by applying the optimal pareto solutions to other summer seasons and another glacier.

[Figure]

The results show that the average parameter uncertainty over the whole glacier is 30% and reaches 50% at point scale, which underlines the significance of using the spatio-temporal cross-validation for model parameterizations. Although transferability tests of the model shows that the performance is worse when applied to the other studied glacier, the magnitude of uncertainty is the same order as the temporal transfer. In conclusion, the authors suggest that the optimized model can be applied to other glacier under similar climatic conditions. The introduction is a very detailed review of the previous studies studying the transferability of different models. I find it really enjoyable to read. And the following sections are well connected with the aim of the article. The method is well explained and is developed from the referenced articles in the last paragraph of introduction. I think the experiment is well designed and written.

The applications of glacier mass and energy balance models in regional upscaling and projections are limited by the high variability of climatic parameters. The authors did a thorough investigation of the sensitivity and uncertainty of models. The proposed two-step process has proved its efficiency in reducing parameter space and identifying the model uncertainties. The identified non-sensitive parameters could be fixed to constant literature values. Then the optimization is performed based on the remaining sensitive parameters. The method is well designed, and the results support the assumption that the single performance measure is not appropriate since parameters are varying in both time and space. The results suggest that the net short-wave radiation and turbulent fluxes are the most uncertain energy balance components, reasserting that the snow and ice albedo representation are the most critical factors. This conclusion explains the difficulty in optimizing the model performance simultaneously in both the accumulation and the ablation zone. Altogether, I regard the paper as a valuable contribution Still, I have a few comments which I hope the authors can address so that I could learn more about their research.

2. General comments

The performance of the model is not that encouraging when applying the 17 optimal

solutions based on the Pareto set for HEF 13 to other summer seasons and another glacier. The conclusion suggests that the large spatial and temporal transfer uncertainties are acceptable when applying to other glaciers with similar climatic settings. How does the result compare to previous research (e.g. the referenced an enhanced temperature-index model by Carenzo et al. (2009) which shows pretty good agreement of transferability in space and time)? The uncertainties of transferability are quantified only through the Euclidean distance towards the utopian point, which is quite clear and straightforward. However, it would have been better if R2 values were also reported, which is helpful for facilitating comparison to earlier transferability studies.

The article has a clear structure with a very thorough description of the parametrization. Some descriptions however need some clarification as specified below in specific comments. The length of the abstract could be shortened by reducing some of the detailed descriptions of the methods.

3. Specific Comments

1) P13, L5:Figure 4 could be improved. It is written that a minor change of a model bias (25Åůkgm-2) could lead to an improvement in MAD by 200-300. However, this statement excludes many outliers which should not be ignored. A log-transform might be able to help to improve

2) P13, L6: "the MADs plane is more curved" in Fig 4(c), (similar statement for line 2 on the same page) seems to be just a vague description. It might help to add a reference line here to support this sentence.

3) P20, L32: A minor typo is spotted where "TFor" is assumed to be "For".

4) Figure 5: the y axis should be "Euclidean" not "euclidian".

5) Figure 6: Maybe it would be good to compare the optimized best setting with the "classical best guess solution" in fig. 6? It's good to have a comparison between the optimal results and the classical best settings. Then the quantified uncertainties

can be compared with previous research? For instance, the transferability of the enhanced temperature-index model (Carenzo et al., 2009), which is reported to have a good transferability (R2 = 0.78 under the over cast conditions and R2 = 0.925 on average under normal conditions). Another study of a distributed energy balance model (MacDougall and Flowers, 2011) concluded that an error of $\sim$30% is expected without calibration during transferability test.

References

Carenzo, M., Pellicciotti, F., Rimkus, S., Burlando, P., 2009. Assessing the transferability and robustness of an enhanced temperature-index glacier-melt model. Journal of Glaciology 55, 258–274. https://doi.org/10.3189/002214309788608804

MacDougall, A.H., Flowers, G.E., 2011. Spatial and temporal transferability of a distributed energy-balance glacier melt model. Journal of Climate 24, 1480–1498. https://doi.org/10.1175/2010JCLI3821.1

---

## Referee Comment (RC2) · Anonymous Referee #2 · 28 Oct 2018

This is a well written and technically adept article describing the application of uncertainty and sensitivity analysis to glacier mass and energy balance models. I enjoyed the paper, but do have three questions regarding the utility and objectives of this study:

1) The paper executes sensitivity and uncertainty analyses of a glacier mass balance model with the goal to "target a clear separation of the concepts of sensitivity and uncertainty". I often struggle with this, because as much as we want these two concepts to be different, they are inherently linked, as they are in your approach to investigate this model. Beyond this, I searched for a clear objective as to why this study was being performed. Why use all of these methods with a single model? What is the targeted

outcome? Why would you encourage others to do the same? More clear statement of these goals upfront and then tying to these goals in the end will help to tie the paper together. In my experience, others don't necessarily see why such a robust and technical approach to modeling is needed - I think you have great fodder to demonstrate why.

2) Many of the figures I struggled to extract the key meaning. In particular, Figure 4, Figure 5, and Figure 6. You might consider, instead, some sort of conceptual figure that aims to bring out your key findings/messages in terms of the sequential application of methods you took. What is learned, and how can you represent this more clearly to others? I enjoyed the other conceptual figures in the manuscript.

3) There is often discussion of the feedback between models and observations. What role does the need for observations play in your study? So much of the discussion was focused on parameters, and I found myself wondering often about the observations.

Minor comments:

-Did you test for convergence in your sensitivity indices? Given the number of model runs, I'm not sure this is needed, but you could get the same results with fewer runs, which might be valuable information for other researchers (and make this type of approach seem more tractable to them)

-It's not clear to me why in Section 3.2.1 why analysis of 10,000 parameter samples is reported, as well as analysis with 300,000 simulations is reported. Why report the 10,000 runs?

-Abstract - line 2 - 'they' is ambiguous

-Figure 3 - consider grouping your parameters by type and using some color or labeling

-Figure 4 - Quite difficult to get anything out of Fig 4(d) – consider making a few more subplots and grouping results, or adding labeling

[Figure]

---

## Author Comment (AC1) · 27 Nov 2018

Tobias Zolles (corresponding author: tobias.zolles@uib.no) Fabien Maussion, Stephan P. Galos, Wolfgang Gurgiser, Lindsey Nicholson

We would like to thank the two referees and the author of the short comment for their effort and their valuable contributions in the discussion of our manuscript. This document starts with a general comment by the authors followed by the replies to the referee comments of Matthieu Lafaysse and the anonymous referee 2, as well as to the short comment by S.Feng. The referee and short comments are given in *italics* while the author replies are presented in regular font.

**1 General comment by the authors**

This study is a technical approach to investigate the sensitivity and uncertainty of a glacier surface energy balance model. Therefore, uncertainty in the meteorological forcing is not considered here. For a total uncertainty quantification of the simulation and projection such an analysis would of course be necessary but it is out of scope of this study. To make this more obvious for the reader we sharpened the text of the revised manuscript where we now put a clearer focus on distinguishing between model sensitivity analysis and model output uncertainty assessment.

**2 Author replies to the comments by referee 1 (Matthieu Lafaysse)**

*The manuscript of Zolles et al presents ensemble simulations of glacier mass balance with an energy balance model applied on 2 glaciers and 3 seasons. The main innovation compared to the existing literature consists in a relatively comprehensive analysis of model sensitivities and uncertainties. The paper is well written and well structured. The statistical framework is clearly explained. I especially appreciate the effort of the authors to give simple examples to explicit the formalism (pages 7; 9). The conclusions are clearly summarized and consistent with the obtained results. The complex equifinality between the parameterizations of such models is well demonstrated and the implication in model calibration and model transferability is very interesting. Therefore,I think this paper deserves publication after a minor revision which would account for my few comments below when it is possible.*

*Page 2 line 19: I think most studies base this statement on an evaluation of the energy fluxes and surface temperature, not only the melt rates.*

We agree and changed the statement according to the referees remark.

*Page 4 line 7: I understand the deficiencies of the cited references but given the number of studies which just present simulation outputs without any uncertainty quantification, I think that the word "inadequate" is a bit severe.*

In the revised manuscript we changed the sentence and replaced "inadequate" by "insufficient".

*Page 4 line 11 / Table 2: the 23 parameters include a "precipitation perturbation" which disappears in the results section (Figure 3) without any specific explanation. More generally, it is not completely clear if the authors want to incorporate the spatialization of meteorological data as part of their model uncertainty study. The decision to exclude the longwave parameterization from the free parameters has a strong consequence in the results. Indeed, large errors are introduced here because Equation A2 is a strong simplification of the real behaviour of the full column of atmosphere. Snow models are usually extremely sensitive to these errors (Sauter and Obleitner, 2015; Quéno et al, 2017). The authors acknowledge this limitation (page 19 lines 5-12) but I do not really understand this choice. Why should the impact of temperature gradient uncertainty on longwave radiation be accounted for if the parameters of equations A3 and A4 are not? Similarly, what is the logic in considering the uncertainty of precipitation gradient but not the uncertainty of the mean precipitation forcing? I think it could also be more explicit in the text that Figure 7 does not represent the full contributions of uncertainties. The very narrow range obtained for longwave radiation is unlikely to represent the real uncertainty of this component as the incoming flux is highly uncertain whereas it is not accounted for.*

As already mentioned in section 1 of this document, the subject of this study is a model internal sensitivity and uncertainty analysis and not an assessment of the absolute model performance. Hence, the uncertainties of the meteorological input parameters are not explicitly considered. The revised manuscript puts a clearer focus on constraining the model sensitivity analysis performed in this study from an output uncertainty assessment. We therefore removed the precipitation perturbation from table 2. We

fully agree that energy balance models are very sensitive to the used parametrizations of longwave radiation but in our study we make use of measured long-wave radiation which is consequently considered as a meteorological input variable. However, in the revised manuscript we discuss this point more clearly.

*Table 2: Can you comment the range of precipitation density? This range is not realistic for snowfall in the Alpine area (too high values, Helfricht et al, 2018). It may compensate some deficiency in a simple model which does not represent accurately compaction but this should be detailed. The authors could also comment the implication in the uncertainty analysis of using some potentially unrealistic values for some parameter ranges. The same could apply if the precipitation perturbation was really considered because a 10% error is not sufficient to represent precipitation uncertainty in mountainous areas.*

The used range of precipitation density is higher than reported by Helfricht et al (2018). By limiting our study to the summer season the effect is lower, but still present. It influences the albedo parameterization through the snow depth scale. This is a shortcoming and based on the new results its range should be increased. The precipitation perturbation would definitely be too low, but it was removed from any simulations to keep the original meteorological input unperturbed. As mentioned before it was not considered anymore in the final simulations. Therefore we removed it also from table 2 . Changes: Besides changing table 2 from which we removed the precipitation perturbation, the revised manuscript explicitly discusses the point of unrealistic parameter ranges, as well as the new findings by Helftricht et al. (2018).

*Page 13 lines 21-22: I am not sure to correctly understand this sentence. Could you develop what you mean by "less constrained" and what is the relationship with a narrow initial range of parameters?*

We changed the text of the revised manuscript according to the suggestion of the referee in order to make this point clear for the reader.

[Figure]

*Page 19 line 1-4: This is true but rather utopic at the moment. Such models need a forcing of impurity depositions. The existing products are not sufficiently reliable nor sufficiently detailed to depict the processes responsible for the spatial variability of albedo on a glacier.*

We rephrased this sentence according to the referee's remark..

*Page 19 lines 13-20: The authors discuss the impact of the possible variability of roughness lengths. However, I think they could also discuss more generally the relevance of applying this theory of turbulent fluxes formulation in mountainous environments where the turbulence is probably more affected by the surrounding topography than by the surface roughness itself (Conway and Cullen, 2013).*

We added a sentence briefly discussing this issue including the citation of Conway and Cullen (2013) and Sauter and Galos (2016).

*Page 19 lines 21-22 Which effects are you talking about? From experiments with a detailed snowpack model (with a sufficient vertical discretization), it is rather clear than the absorption profile has an impact on surface temperature and on the temperature gradient close to the surface (and therefore on snow metamorphism). However, the effect on more integrated variables is likely to be much less significant.*

The statement was removed.

*Page 20 line 15 I do not know whether new field experiments on that topic are really required right now. The authors should first mention that the relationship between albedo and grain shapes and sizes is already implemented in detailed snowpack models such as Crocus (Vionnet et al, 2012) or SNOWPACK (Lehning et al, 2002).*

We agree that those models have a better parametrization for the snow albedo. The statement was adjusted and the references added.

*Page 20 lines 30-32 I agree and the same applies for various variables, especially surface temperature which is a good indicator of the correct resolution of the energy*

*balance.*

Thanks.

*Page 20 line 33 The lack of a full quantification of the meteorological uncertainty is probably the main limitation of this paper. This is only stated here in a small paragraph which would have deserved to be more developed based on the existing literature. Indeed, this is probably the most studied uncertainty in previous studies in snow modelling (e.g. Raleigh et al, 2015) and in glacier modelling. However, the possible compensation errors between meteorological forcing and model parameters may deteriorate the relevance of model uncertainty studies which do not account for forcing uncertainties. I did the same thing myself in the context of a detailed multiphysics snowpack modelling (Lafaysse et al, 2017) but I just think that this limitation could be more discussed.*

We agree. Besides the changes presented above (c.f. sect. 1 of this document) the revised manuscript contains a more explicit discussion of this issue.

*Page 21 line 27: To what does 1 $kgm^{-2}$ refer? In which duration?*

Thank you for spotting this error. The true value is 1000 $kgm^{-2}$ per summer season. We corrected the value and clarified the statement.

*Typos: Abstract line 5: "which" introduces Page 16 line 26: energy melt energy Page 19 line 4: change Page 21 line 32: For*

We corrected all the typos indicated by the referee.

**3   Author replies to the comments by referee 2 (anonymous)**

*The paper executes sensitivity and uncertainty analyses of a glacier mass balance model with the goal to "target a clear separation of the concepts of sensitivity and un-*

*certainty". I often struggle with this, because as much as we want these two concepts to be different, they are inherently linked, as they are in your approach to investigate this model. Beyond this, I searched for a clear objective as to why this study was being performed.*

We agree that the two concepts cannot be regarded as fully independent and it is therefore not always trivial to provide a clear separation. As already mentioned in our general comment in Sect.1 we put a much stronger emphasis on this issue in the revised manuscript. Besides that we provide a more explicit motivation for our study.

*Why use all of these methods with a single model? What is the targeted outcome? Why would you encourage others to do the same? More clear statement of these goals upfront and then trying to these goals in the end will help to tie the paper together. In my experience, others don't necessarily see why such a robust and technical approach to modeling is needed - I think you have great fodder to demonstrate why.*

We agree that an application of a similar method to more models would be highly appreciated. This study was limited to one model to keep it simple enough and have a clear focus on the technical details. In our study we decided to focus on the fundamental question: How robust is a single "best guess"/optimal solution? Our study clearly shows that a single solution is not representative despite providing good results within the calibration period. Additionally, the insights by our study may reduce computational costs in future studies as parameters with a low sensitivity may be kept fixed. However, we applied several changes to the revised manuscript to make the goal of our study more obvious and in particular address the topics of parameter overfitting and the representativeness of single best guess solutions.

*Many of the figures I struggled to extract the key meaning. In particular, Figure 4, Figure 5, and Figure 6. You might consider, instead, some sort of conceptual figure that aims to bring out your key findings/messages in terms of the sequential application of methods you took. What is learned, and how can you represent this more clearly to*

*others? I enjoyed the other conceptual figures in the manuscript.*

We thank the referee for the suggestion of a conceptual figure and decided to go for a flow chart like image to explain the sequential approach. Such a figure is well suited in the end of the introduction to explain our aims goals and sequential approach prior to explaining the details but, due to its placing within the manuscript the results are not included in the figure. For this reason we decided to also keep the other figures. However, we improved the figure captions and added a clearer explanation where this was necessary. Figure 5 was moved to the supplement. The typo in figure 6 (Euclidean) was corrected.

*There is often discussion of the feedback between models and observations. What role does the need for observations play in your study? So much of the discussion was focused on parameters, and I found myself wondering often about the observations.*

There are two parts of observations to consider: 1. the meteorological input 2. the data used in the optimization process

1. Again we would like to clarify that our study focuses on model sensitivities and does hence not deal with uncertainties in the forcing data (see general comment in sect. 1 of this document and comments above). We put stronger emphasis on this issue in the revised manuscript. 2. The uncertainty in the mass balance data is discussed in more detail in the revised manuscript than it was in the original one. The revised manuscript also expands the discussion on other objective functions based on additional observations.

Minor referee comments

*Did you test for convergence in your sensitivity indices? Given the number of model runs, I'm not sure this is needed, but you could get the same results with fewer runs, which might be valuable information for other researchers (and make this type of approach seem more tractable to them)*

We did not use an evolutionary setup to test for convergence. The first approach used 25.000 runs for the GSA, which was enough for confidence in the accumulation area, but not in the ablation zone. Instead of continuously increasing we just changed it to a base sample of 12.000, leading to a total of 300.000 runs for the GSA. The total cost of simulations is the N*(k+2) with N the base sample and k the number of parameters. In general the base sample and total ensemble can be continuously increased in size if necessary until convergence is achieved. Bootstrapping in this approach leads to the estimation of the sensitivity indices. The convergence criteria that were used here: $S_{xi} \leq S_{Ti}, \sum S_{Xi} < 1$ (Saltelli, 2000; 2006; 2010). Finally, we required that the variance of the sensitivity indices after bootstrapping does not interfere with our sensitivity criteria of $T_{Si}$<0.05. If only the mean of the sensitivity index is considered the 25.000 runs already show the same result, but with a lower confidence as close the ELA for sensitivity of individual parameters is quite uncertain at this number. We included a mathematical description of the quality assessment of the method in the revised manuscript and that the performance of fewer solutions was not investigated.

*It's not clear to me why in Section 3.2.1 why analysis of 10,000 parameter samples is reported, as well as analysis with 300,000 simulations is reported. Why report the 10,000 runs?*

The 10.000 runs are for the exemplary simulation of the simple model $Y = X_{12} + X_3$ The sample size i irrelevant for the intention of this simple model and therefore removed the statement in the revised manuscript.

*Abstract - line 2 - 'they' is ambiguous*

The sentence was changed to avoid ambiguity.

*Figure 3 - consider grouping your parameters by type and using some color or labeling*

We changed the order of the parameters though to make it clearer and grouped the momentum roughness length of fresh snow with the other turbulent flux related parameters. The revised figure has a clear grouping of the parameters However, some parameters have a common type (for example fresh snow/firn/ice albedo) and influence directly the same quantity (surface albedo). Therefore we already grouped them in the initial order, but without a clear separation. This was intentional as for example the fresh snow density does influence the albedo, as well as subsurface process.

*Figure 4 - Quite difficult to get anything out of Fig 4(d) – consider making a few more subplots and grouping results, or adding labeling*

We are aware of the difficulty of reading this subplot, but this is the main finding: There is not really a big feature to observe. We added a better description of this subplot to the figure caption and the manuscript.

**4   Author replies to the short comment by S. Feng**

General comments

*The performance of the model is not that encouraging when applying the 17 optimal solutions based on the Pareto set for HEF 13 to other summer seasons and another glacier. The conclusion suggests that the large spatial and temporal transfer uncertainties are acceptable when applying to other glaciers with similar climatic settings. How does the result compare to previous research (e.g. the referenced an enhanced temperature-index model by Carenzo et al. (2009) which shows pretty good agreement of transferability in space and time)? The uncertainties of transferability are quantified only through the Euclidean distance towards the utopian point, which is quite clear and straightforward. However, it would have been better if $R2$ values were also reported, which is helpful for facilitating comparison to earlier transferability studies.*

Indeed the spatial and temporal transfer of the optimal solutions are not particularly encouraging. Although the settings may be transferable between certain cases, this does

not generally hold. Te order of magnitude of the maximum transfer errors is similar for time and space To put this into the context of previous studies we want to briefly comment on the following points: First the general performance based on our criteria (MAD, RMSD) is worse than to the reference (energy balance model) in Carenzo et al. (2009), but it compares to different quantities: differences between two models and differences between measurements and model. The model performance over a variety of points relative to the measurements may not be that great. We observe a similar possibility in our tuning that the cumulative mass balance, which our bias over the stakes functioned as a proxy, is easier to minimize than the other two criteria. Both energy balance and the enhanced temperature index may have similar biases. The spatial transfer is further worse for our model as we do calculate the solar radiation (based on cloud cover deductions from one weather station) and the albedo. Also Carenzo et al. (2009) find a worse transferability in the case of calculated solar radiation. Furthermore, additional uncertainty is introduced for us as also temperature and precipitation are downscaled values from one station. Furthermore, our study has 6 members with distinct variations in mass balances ranging from drastically negative to positive while the total UDG in the Carenzo study varies from 4300-3200kg/m$^2$ being clearly dominated by stakes with more negative mass balance. We did not include R$^2$ as it is a much weaker statistical measure than the multi-objective approach used here. The MAD/RMSD serve a similar purpose. However, R$^2$ is not a good measure for model performance. This is especially true if different data sets and models are compared. Furthermore, there is additional variance in our mass balance measurement data (avalanche, snow redistribution,...) which lowers R$^2$, while this effect is not present if you compare model to model as done in Carenzo et al. (2009). For more details we refer to Shalzi (2015) and in Berk (2004). However, in the revised manuscript we provide a better explanation why the particular objective functions were used in this study .

*The article has a clear structure with a very thorough description of the parametrization. Some descriptions however need some clarification as specified below in specific comments. The length of the abstract could be shortened by reducing some of the*

[Figure]

*detailed descriptions of the methods.*

Specific Comments

*P13, L5:Figure 4 could be improved. It is written that a minor change of a model bias could lead to an improvement in MAD by 200-300. However, this statement excludes many outliers which should not be ignored. A log-transform might be able to help to improve.*

It is not fully clear to the authors what is meant with "outliers". There are no outliers in a Pareto-set. We specifically did choose not to use a log plot to have similar scales which enables the reader to see the difference in performance on a graspable scale (in kg/m$^2$, added for clarification to the figure).

*P13, L6: "the MADs plane is more curved" in Fig 4(c), (similar statement for line 2 on the same page seems to be just a vague description. It might help to add a reference line here to support this sentence.*

We changed the sentence and added an explanation.

*P20, L32: A minor typo is spotted where "TFor" is assumed to be "For".*

We corrected the typo.

*Figure 5: the y axis should be "Euclidean" not "euclidian".*

The axis label was capitalized in the revised manuscript in all figures.

*Figure 6: Maybe it would be good to compare the optimized best setting with the "classical best guess solution" in fig. 6? It's good to have a comparison between the optimal results and the classical best settings. Then the quantified uncertainties or instance, the transferability of the enhanced temperature-index model (Carenzo et al., 2009), which is reported to have a good transferability (R2 = 0.78 under the overcast conditions and R2 = 0.925 on average under normal conditions). Another study of a distributed energy balance model (MacDougall and Flowers, 2011) concluded that an*
*error of â$\acute{L}$ij30% is expected without calibration during transferability test.*

The question is what is considered as the classical best guess. There is a diverse variation in literature with the MAD, RMSD, $R^2$ relative to individual or multiple stake or glacier wide mass balance. This study treats the optimization as an ensemble with emphasizing the issues of best guess scenarios. Nevertheless, that is exactly what figure 6 shows. The compromise solution is our best guess. We included the classical best guess relative to the MADs and BIAS in figure 5 to allow for a comparison of the optimal solution space and the classical best guess settings and our best guess. However, the figure was moved to the supplement. We want to emphasize once more that a single best guess is of limited used and put a stronger emphasize on this in the manuscript. MacDougall and Flowers (2011) report a spatial transfer error of up to 530 mm w.e. (kgm$-2$). They furthermore report larger errors in the ablation zone. We attribute thelarger uncertainties in our study to a larger variation in measured mass balance over the sample period, a larger distance between the glaciers and the upper limit estimation based on the multi-objective approach. We added a discussion of these points to the revised manuscript and we put our results in context to the above mentioned studies.

**5  References**

Berk, Richard A. Regression analysis: A constructive critique. Vol. 11. Sage, 2004.

Carenzo, M., Pellicciotti, F., Rimkus, S., Burlando, P., 2009. Assessing the transferability and robustness of an enhanced temperature-index glacier-melt model. Journal of Glaciology 55, 258–274. https://doi.org/10.3189/002214309788608804

Conway, J. and Cullen, N.: Constraining turbulent heat flux parameterization over a temperate maritime glacier in New Zealand, Annals of Glaciol., 54, 41-51, doi:10.3189/2013AoG63A604, 2013

Helfricht, K., Hartl, L., Koch, R., Marty, C., Olefs, M.: Obtaining sub-daily new snow density from automated measurements in high mountain regions, Hydrol. Earth Syst. Sci., 22, 2655-2668, doi:10.5194/hess-22-2655-2018, 2018

Lafaysse, M., Cluzet, B., Dumont, M., Lejeune, Y., Vionnet, V. and Morin, S.: A multiphysical ensemble system of numerical snow modelling, The Cryosphere, 11, 1173-1198, doi:10.5194/tc-11-1173-2017, 2017

MacDougall, A.H., Flowers, G.E., 2011. Spatial and temporal transferability of a distributed energy-balance glacier melt model. Journal of Climate 24, 1480–1498. https://doi.org/10.1175/2010JCLI3821.1

Klug, C., Bollmann, E., Galos, S. P., Nicholson, L., Prinz, R., Rieg, L., Sailer, R., Stötter, J., and Kaser, G.: A reanalysis of one decade of 10 the mass balance series on Hintereisferner, Ötztal Alps, Austria: a detailed view into annual geodetic and glaciological observations, The Cryosphere Discussions, 1976, 1–38, doi:10.5194/tc-2017-132, 2017

Raleigh, M. S., Lundquist, J. D., and Clark, M. P.: Exploring the impact of forcing error characteristics on physically based snow simulations within a global sensitivity analysis framework, Hydrol. Earth Syst. Sci., 19, 3153–3179, doi:10.5194/hess-19-3153- 2015,2015.

Rye, C. J., Willis, I. C., Arnold, N. S., and Kohler, J.: On the need for automated multiobjective optimization and uncertainty estimation of glacier mass balance models, Journal of Geophysical Research, 117, 1–21, doi:10.1029/2011JF002184, 2012

Saltelli, A., Campolongo, F., and Tarantola, S.: Sensitivity Analysis as an Ingredient of Modeling, Statistical Science, 15, 377–395, 20 doi:10.1214/ss/1009213004, 2000.

Saltelli, A., Ratto, M., Tarantola, S., and Campolongo, F.: Sensitivity analysis practices: Strategies for model-based inference, Reliability Engineering and System Safety, 91, 1109–1125, doi:10.1016/j.ress.2005.11.014, 2006.

[Figure]

Saltelli, A., Annoni, P., Azzini, I., Campolongo, F., Ratto, M., and Tarantola, S.: Variance based sensitivity analysis of model output. Design and estimator for the total sensitivity index, Computer Physics Communications, 181, 259–270, doi:10.1016/j.cpc.2009.09.018, 2010.

Sauter, T. and Galos, S. P.: Effects of local advection on the spatial sensible heat flux variation on a mountain glacier, Cryosphere, 10, 2887–2905, doi:10.5194/tc-10-2887-2016, 2016.

Sauter, T. and Obleitner, F.: Assessing the uncertainty of glacier mass-balance simulations in the European Arctic based on variance decomposition, Geosci. Model Dev., 8, 3911–3928, doi:10.5194/gmd-8-3911-2015, 2015.

Shalzi, C, Lecture 10:F-Tests,R2, and Other Distractions, 2015, downloaded from http://www.stat.cmu.edu/ cshalizi/mreg/15/lectures/10/lecture-10.pdf 5.11.2018

Vionnet, V., Brun, E., Morin, S., Boone, A., Faroux, S., Le Moigne, P., Martin, E.,and Willemet, J.-M.: The detailed snowpack scheme Crocus and its implementation in SURFEX v7.2, Geosci. Model Dev., 5, 773–791, doi:10.5194/gmd-5-773-2012, 2012

Zemp, M., Thibert, E., Huss, M., Stumm, D., Rolstad Denby, C., Nuth, C., Nussbaumer, S. U., Moholdt, G., Mercer, A., Mayer, C., Joerg, P. C., Jansson, P., Hynek, B., Fischer, A., Escher-Vetter, H., Elvehøy, H., and Andreassen, L. M.: Reanalysing glacier mass balance measurement series, The Cryosphere, 7, 1227–1245, doi:10.5194/tc-7-1227-2013, 2013.